# Many-body chaos near a thermal phase transition

**Alexander Schuckert[1,2][*] and Michael Knap[1,2]**

**1** Department of Physics and Institute for Advanced Study,
Technical University of Munich, 85748 Garching, Germany
**2** Munich Center for Quantum Science and Technology (MCQST),
Schellingstr. 4, D-80799 München

[*] alexander.schuckert@tum.de

## Abstract

We study many-body chaos in a (2+1)D relativistic scalar field theory at high temperatures in the classical statistical approximation, which captures the quantum critical regime and the thermal phase transition from an ordered to a disordered phase. We evaluate out-of-time ordered correlation functions (OTOCs) and find that the associated Lyapunov exponent increases linearly with temperature in the quantum critical regime, and approaches the non-interacting limit algebraically in terms of a fluctuation parameter. OTOCs spread ballistically in all regimes, also at the thermal phase transition, where the butterfly velocity is maximal. Our work contributes to the understanding of the relation between quantum and classical many-body chaos and our method can be applied to other field theories dominated by classical modes at long wavelengths.

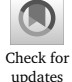
# 1   Introduction

Thermalization in classical many-body systems can be understood from the perspective of dynamical chaos: details of the initial state are effectively forgotten by the exponential divergence of trajectories. In quantum many-body systems, the same picture can not be immediately applied as the Schrödinger equation is a linear differential equation and therefore does not directly give rise to chaos. However, parts of the system may look thermal if their surrounding provides a thermalizing environment, as put forward by the eigenstate thermalization hypothesis [1–3]. Thermalization can be directly probed by evaluating fluctuation-dissipation relations far from equilibrium as well [4–6]. Yet, a dynamical mechanism of thermalization in quantum systems comparable in generality to the one offered by chaos in classical systems has remained elusive so far.

Recently, "out-of-time-ordered correlation functions" (OTOCs) [7,8] have been proposed as a generalization of classical dynamical chaos to quantum systems. As motivated from the perspective of operator scrambling in strongly coupled field theories with a gravity dual [9–13], they have been shown to exhibit exponential growth in many field theories [14–20]. Moreover, OTOCs spread (in general) ballistically in space with a "butterfly velocity" quantifying the speed of scrambling, which has been also found in non-relativistic lattice systems [21–30]. Such ballistic spreading is to be expected in systems with well-defined quasiparticles [31], but even strongly coupled systems without quasiparticles exhibit a well-defined butterfly velocity. While these results show many analogies to dynamical chaos in classical systems [32–35], the exact relation between exponential growth in OTOCs and classical chaos remains unclear.

Here, we study a self-interacting real scalar field theory in the strongly correlated regime in which classical modes are expected to dominate: at high temperatures and around a second-order thermal phase transition. While the critical dynamics are notoriously hard to study with diagrammatic techniques [36], the classical statistical approximation provides reliable results for the order parameter dynamics in these regimes [37–41]. Furthermore, it has recently been shown that also the leading behaviour of the OTOC is captured within semi-classical approximations [42] and we conjecture this result to generalize to our case. Hence, we numerically obtain both the spectral function and the OTOC by evolving the classical field equations of motion of an infinitesimal perturbation and averaging over thermal initial states. The zero-momentum spectral function exhibits algebraically slow relaxation near the critical point as a consequence of critical slowing down, but possesses well-defined quasi-particles at higher momentum or away from the critical point. By contrast, we do not find signatures of critical slowing down in the OTOC even though the studied time scales are well within the temporal correlation length. Instead, the OTOC exhibits ballistic spreading and exponential growth in the whole considered parameter regime. By matching the quantum field theory in the quantum critical regime to the classical field theory via dimensional reduction, we find the Lyapunov exponent to reproduce the linear-in-temperature scaling, that has been found in other strongly coupled theories in accordance with the Maldacena-Shenker-Stanford (MSS) bound [10]. Furthermore, it approaches the non-interacting limit algebraically in a fluctuation parameter and

exhibits a cusp at the phase transition. The butterfly velocity is significantly smaller than the speed of light and shows a global maximum near the phase transition. Lastly, the temporal fluctuations of the OTOC follow a self-similar behaviour in agreement with the Kardar-Parisi-Zhang (KPZ) universality class [43].

This work is organized as follows. First, we introduce the real scalar field theory, how to obtain the classical statistical approximation from dimensional reduction and our numerical methods. Secondly, we discuss the long wavelength excitations obtained from the spectral function at zero momentum. Finally, we show that the dynamics of the OTOC offer a qualitatively different perspective on the thermalization dynamics compared to the spectral function.

## 2 Real scalar field theory at high temperature

**Model.** We study a real scalar field theory in $d = 2$ spatial dimensions given by the Hamiltonian

$$H = \int d^2\mathbf{x} \left[ \frac{1}{2}\pi^2 + \frac{1}{2}(\nabla\varphi)^2 + \frac{1}{2}m^2\varphi^2 + \frac{\lambda}{4!}\varphi^4 \right], \tag{1}$$

with bare mass $m^2$ and interaction constant $\lambda$. $\pi = \partial_t\varphi$ is the canonically conjugate momentum of the real scalar fields $\varphi$. This model exhibits a finite temperature phase transition from a disordered paramagnetic phase with $\langle\varphi\rangle = 0$ to a symmetry broken phase with $\langle\varphi\rangle \neq 0$ in the universality class of the 2D Ising model.

**The classical statistcal approximation.** At high temperatures and close to the phase transition, the two dimensional classical statistical field theory given by the Hamiltonian in Eq. (1) can be interpreted as an effective field theory for the corresponding (2+1)D finite temperature quantum field theory for long-wavelength, long-distance properties. This may be allegorically understood from the fact that in the high-T regime and close to a thermal phase transition, dominant long wavelength excitations have frequency $\omega \ll T$ and hence the Bose-Einstein distribution reduces to the classical Rayleigh-Jeans law

$$\frac{1}{\exp(\omega/T) - 1} \approx \frac{T}{\omega} \gg 1. \tag{2}$$

For long wavelength observables, the quantum field theory is therefore expected to be dominated by these highly occupied modes and reduces to a classical statistical field theory (see appendix B for a discussion of this dynamical argument). The two dimensional classical theory may then be matched to the corresponding (2+1) dimensional quantum field theory by inserting the mass $m^2$ and coupling $\lambda$ obtained from dimensional reduction, i.e., by integrating out all non-zero Matsubara frequencies. This procedure is based on the fact that the latter have larger thermal masses than the zero Matsubara mode to lowest order in an $\epsilon = 3 - d$ expansion [41] and hence may be integrated out to obtain the long-distance properties of the theory (see appendix A for a short summary of dimensional reduction). While it was previously shown that this procedure can also be used to obtain the order parameter dynamics [41, 44, 45], we conjecture in this work that it also captures the leading chaotic scrambling dynamics in the OTOC at times shorter than the Ehrenfest time. This assumption was previously shown to be valid in semi-classical calculations in the Bose-Hubbard model [42]. Furthermore, a diagrammatic approach to the related O(N) model has found the OTOC to be dominated by momenta $p < T$, i.e., the classical modes at high temperature [16].

**Observables at finite temperature.** We evaluate all observables $\mathcal{O}$ in thermal equilibrium according to the classical phase space average

$$\langle \mathcal{O}(\mathbf{x}, t) \rangle_{\text{cl}} = \frac{1}{Z_{\text{cl}}} \int \mathcal{D}\varphi_0 \mathcal{D}\pi_0 \mathcal{O}(\mathbf{x}, t) \exp(-H/T), \tag{3}$$

where $Z_{\text{cl}} = \int \mathcal{D}\varphi_0 \mathcal{D}\pi_0 \exp\{-H/T\}$ is the classical partition sum at temperature $T$ and the phase space measure at the initial time is given by $\mathcal{D}\varphi_0 \mathcal{D}\pi_0 = \Pi_{\mathbf{x}} d\varphi(\mathbf{x}, t = 0) d\pi(\mathbf{x}, t = 0)$. Numerically, we regularize the model on an $N \times N$ lattice with lattice spacing $a_s$ and sample the canonical distribution with a hybrid Monte Carlo method. For details on the implementation, see appendix C.

All ultraviolet divergences of the classical field theory are cancelled after lattice regularization [46] by introducing a one-loop renormalized mass[1] $M$ according to

$$m^2 = M^2 - \frac{\lambda T}{2} \int \frac{d^2 \mathbf{p}}{(2\pi)^2} \frac{1}{\mathbf{p}^2 + M^2}, \tag{4}$$

which is the classical limit of the corresponding result in thermal quantum field theory [38,41].

Furthermore, we use $M$ as our unit and introduce dimensionless variables according to $\tilde{\mathbf{x}} = \mathbf{x}M, \tilde{t} = tM, \tilde{\varphi}_a = \varphi_a T^{-1/2}, \tilde{\pi}_a = \pi_a M^{-1} T^{-1/2}$. As a result, the theory only depends on a single dimensionless variable

$$\mathcal{G} = \frac{\lambda T}{M^2} \tag{5}$$

and results in the continuum, infinite volume limit are obtained by taking $a_s \to 0, N \to \infty$.

The fluctuation parameter $\mathcal{G}$ interpolates smoothly between the paramagnetic phase (for small $\mathcal{G}$), the high-T quantum critical regime (around $\mathcal{G} \approx 35$) [41], the finite temperature phase transition line at $\mathcal{G}_c \approx 61.44$,[2] and the symmetry broken phase for $\mathcal{G} > \mathcal{G}_c$. We confirmed this value in our Monte Carlo simulations by studying both the Binder cumulant, a measure for non-Gaussian fluctuations of the order parameter (see App. D), and critical behaviour of the spectral function near $\mathcal{G}_c$ in section 3.

**Dynamical correlation functions.** The dynamics of the field can be obtained from its equation of motion

$$\partial_t^2 \tilde{\varphi} = \Delta \tilde{\varphi} - \frac{m^2}{M^2} \tilde{\varphi} - \frac{\mathcal{G}}{6} \tilde{\varphi}^3 \tag{6}$$

on the lattice, with initial conditions obtained from Monte Carlo sampling. In the classical limit, the spectral function $\rho_q(t, \mathbf{x}) = i\langle [\hat{\varphi}(\mathbf{x}, t), \hat{\varphi}(\mathbf{0}, 0)] \rangle$ is given in terms of the Poisson bracket according to $\rho(t, \mathbf{x}) = -\langle \{\varphi(\mathbf{x}, t), \varphi(\mathbf{0}, 0)\}_{PB} \rangle_{\text{cl}}$ and can be obtained from the two-point correlation function using the classical fluctuation-dissipation relation [37,38,41] (FDR)

$$\rho(t, \mathbf{x}) = -\frac{1}{T} \partial_t \langle \varphi(t, \mathbf{x})\varphi(0, \mathbf{0}) \rangle_{\text{cl}}. \tag{7}$$

The spectral function can also be directly obtained from the Poisson bracket (PB),

$$\rho(\mathbf{x}, t) = -\int d^d \mathbf{z} \left\langle \left( \frac{\delta \varphi(\mathbf{x}, t)}{\delta \varphi(\mathbf{z}, 0)} \frac{\delta \varphi(\mathbf{0}, 0)}{\delta \pi(\mathbf{z}, 0)} - \frac{\delta \varphi(\mathbf{x}, t)}{\delta \pi(\mathbf{z}, 0)} \frac{\delta \varphi(\mathbf{0}, 0)}{\delta \varphi(\mathbf{z}, 0)} \right) \right\rangle_{\text{cl}} = \left\langle \frac{\delta \varphi(\mathbf{x}, t)}{\delta \pi(\mathbf{0}, 0)} \right\rangle_{\text{cl}}. \tag{8}$$

---

[1]$M$ is in general not the *full* renormalized mass, in particular it does not vanish at the phase transition. In the perturbative regime at small coupling and high temperatures (small $\mathcal{G}$), $M$ is however very close to the full renormalized mass as pointed out in [46] and shown in section 3.

[2]Note that this value is independent of the cut-off for the latter being small enough as we have removed all ultraviolet divergences with our renormalization procedure [47]. See Ref. [41] for a discussion of the behaviour of $\mathcal{G}$ across the phase diagram.

The latter functional derivative can be evaluated numerically by evolving the linearized field equations of motion in parallel,

$$\partial_t^2 \delta\tilde{\varphi} = \Delta\delta\tilde{\varphi} - \frac{m^2}{M^2}\delta\tilde{\varphi} - \frac{\mathcal{G}}{2}\tilde{\varphi}^2\delta\tilde{\varphi}. \tag{9}$$

This procedure is equivalent to evolving a second field configuration with slightly perturbed initial momenta $\tilde{\pi}(0,\mathbf{x}) \to \tilde{\pi}(0,\mathbf{x}) + \epsilon\delta\tilde{\pi}(0,\mathbf{x})$ in the limit $\epsilon \to 0$. In appendix C we show that obtaining the spectral function numerically with Eq. (8) is equivalent to Eq. (7). In fact, Eq. (8) has some advantages over the FDR method as it does not show finite time-step pathologies for short times. Moreover, it can be used to evaluate the spectral function in regimes in which the FDR does not hold, such as for out-of-equilibrium initial states, as it is related to the numerical linear response theory introduced in Ref. [48, 49].

**Out-of-time ordered correlator (OTOC).** The classical limit of the out-of-time-ordered correlator $-\langle [\hat{\varphi}(\mathbf{x},t), \hat{\varphi}(\mathbf{0},0)]^2\rangle$ can similarly be obtained by replacing commutators with Poisson brackets, giving

$$C(\mathbf{x},t) = \left\langle \{\varphi(\mathbf{x},t),\varphi(\mathbf{0},0)\}_{PB}^2 \right\rangle_{\mathrm{cl}} = \left\langle \left( \frac{\delta\varphi(\mathbf{x},t)}{\delta\pi(\mathbf{0},0)} \right)^2 \right\rangle_{\mathrm{cl}}. \tag{10}$$

Hence, the only difference to the evaluation of the spectral function in Eq. (8) is to square the Poisson bracket *before* averaging over the thermal initial conditions. This means that fluctuations between individual realizations do not cancel out and the chaotic growth of initial perturbations is revealed.

In our simulations, we initialize the perturbation of the momentum as $\delta\pi(\mathbf{x},0) = c\delta(\mathbf{x})$ with a random number $c$ uniformly drawn from a small interval centred around zero, while the field perturbation vanishes initially. Subsequently, we observe the growth of the latter by evolving Eqs. (6,9) in parallel. We found that choosing an initial condition with $\delta\pi(\mathbf{x},0) = 0$, $\delta\varphi(\mathbf{x},0) = c\delta(\mathbf{x})$ as initial condition, i.e. the OTOC $\langle \{\varphi(\mathbf{x},t),\pi(\mathbf{0},0)\}^2 \rangle$, gives similar results as the one in Eq. (10).

## 3 Quasiparticles and critical behaviour in the spectral function

Many properties of a many-body system can be deduced from the nature of its elementary excitations and one might expect scrambling and the spreading of OTOCs to be primarily determined by their properties. We discuss in the next section that this is in fact not the case. Before doing so, we study the spectral function, discussing the qualitatively different regimes for small $\mathcal{G}$ and near the phase transition at $\mathcal{G}_c$.

**Well defined quasiparticles at small** $\mathcal{G}$**.** In weakly interacting theories and in the absence of instabilities, the effective low-energy excitations are generically given by quasiparticles with masses and lifetimes modified by interactions. In real scalar field theory described by the Hamiltonian in Eq. (1), free relativistic bosonic excitations with dispersion $\omega(\mathbf{p}) = \sqrt{\mathbf{p}^2 + m^2}$ are dressed by classical statistical (thermal) fluctuations[3]. To make this statement more explicit, consider the spectral function $\rho(\omega,\mathbf{p})$, which can be written as [38, 50]

$$\rho(\omega,\mathbf{p}) = \frac{-2\,\mathrm{Im}\Sigma(\omega,\mathbf{p})}{[\omega^2 + \mathbf{p}^2 + m^2 - \mathrm{Re}\Sigma(\omega,\mathbf{p})]^2 + [\mathrm{Im}\Sigma(\omega,\mathbf{p})]^2}, \tag{11}$$

---

[3]In the dimensionally reduced theory, quantum fluctuations only enter through the effective couplings $M$ and $\mathcal{G}$ chosen in the classical Hamiltonian.

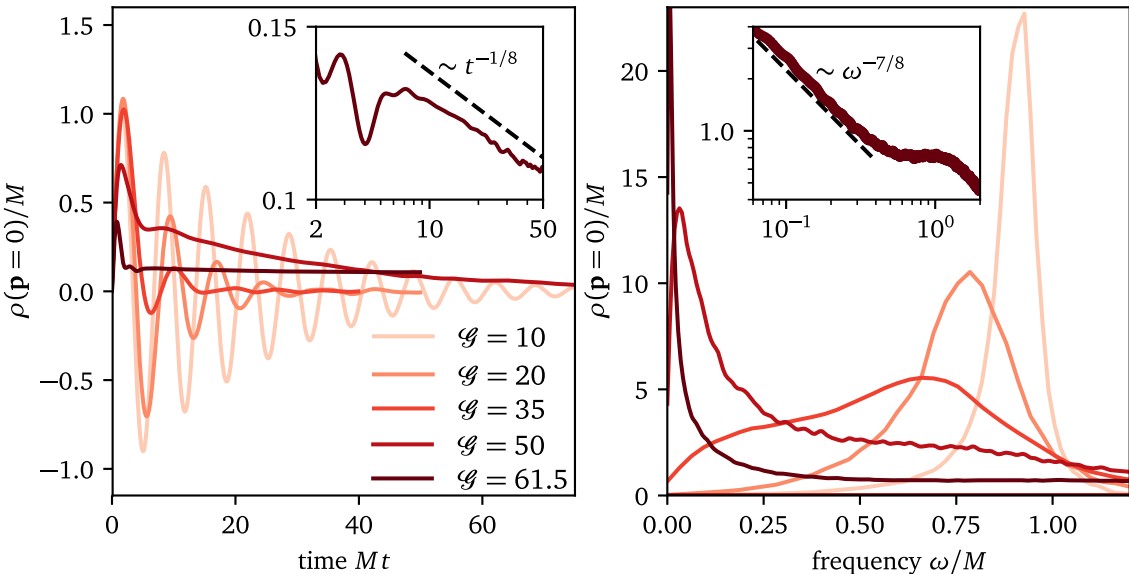

Figure 1: **Spectral function in real time (left) and frequency (right).** For small $\mathcal{G}$, the real time spectral function exhibits weakly damped oscillations, corresponding to a sharp quasiparticle peak with mass gap approximately given by the one-loop renormalized mass $M$. As $\mathcal{G}$ is increased, higher loop corrections become important, leading to stronger damping and a shift of the maximum away from $M$. At $\mathcal{G} = 50$, the spectral function already exhibits critical behaviour, showing an exponential decay with correlation length $\xi_t \approx 30M$. Close to the phase transition, $\mathcal{G} = 61.5$, critical algebraic decay becomes apparent both in time and frequency space as $\xi_t$ is larger than the studied timescales (insets). Dashed lines correspond to the expectations from the 2D static Ising universality class and assuming a dynamic critical exponent $z = 2$. In this plot, $N = 128, a = 0.2$ except for $\mathcal{G} = 61.4$, where $N = 256$. Some regimes of this figure have been previously studied in Refs. [37, 38, 41].

in terms of the (retarded) self-energy $\Sigma(\omega, \mathbf{p})$.

When $\Sigma \to 0$, the spectral function exhibits $\delta$ peaks at the free particle excitation energies $\pm\sqrt{\mathbf{p}^2 + m^2}$. When $\Sigma \neq 0$, sharp peaks still dominate the spectral function as long as the damping rate $-\mathrm{Im}\Sigma(\omega, \mathbf{p})/\omega$ is much smaller than $\mathbf{p}^2 + m^2 - \mathrm{Re}\Sigma(\omega, \mathbf{p})$. The spectral function can then be approximated by a relativistic Breit-Wigner function, reading at zero momentum,

$$\rho_{QP}(\omega, \mathbf{p} = \mathbf{0}) = \frac{2\omega\Gamma}{(\omega^2 - m_R^2)^2 + \omega^2\Gamma^2}, \tag{12}$$

where $m_R^2 = m^2 + \mathrm{Re}\Sigma$ is the renormalized mass of the quasiparticle (QP) state and $\Gamma = -\mathrm{Im}\Sigma/m_R$ is its inverse lifetime.

In Fig. 1 we display the real-time and real-frequency spectral function obtained from the classical FDR in Eq. 7. For $\mathcal{G} = 10$, exponentially damped oscillations in the real-time domain correspond to a quasiparticle peak with Breit-Wigner line shape in the frequency domain, with oscillation frequency and damping rate corresponding to the position and width of the peak, respectively. Fitting the line shape in Eq. (12) to the data, we determine the mass of the quasiparticle as $m_R \approx 0.91M$ and the damping rate (inverse lifetime) as $\Gamma \approx 0.09/M$. By studying the spectral function with $\mathbf{p} > 0$ (not shown in plot) we furthermore find that the effective dispersion of the quasiparticles is given by $\approx \sqrt{m_R^2 + \mathbf{p}^2}$, i.e. the momentum dependence of the effective mass can be approximately neglected in this regime. However, both the height of the peak and the damping rate becomes smaller as $\mathbf{p}$ is increased, which is primarily a result

of the sum rule $\int_0^\infty d\omega/(2\pi)\omega\rho(\omega,\mathbf{p}) = 1$ (i.e. the equal-time commutation relations) as discussed in Ref. [50]. We can conclude that in this regime, well-defined quasiparticles are present, with self-energy corrections beyond the one-loop contribution in Eq. (4) only playing a minor role. For $\mathcal{G} = 20$, the quasiparticle peak broadens and shifts to smaller frequencies as higher-loop corrections become more important, with fit parameters resulting as $m_R \approx 0.78M$ and $\Gamma \approx 0.24/M$. On the $N = 100$ lattice, we find the height of the quasiparticle peak at the first nonzero lattice momentum (not shown in plot) to be already an order of magnitude smaller. This shows that the field dynamics are dominated by small momentum excitations.

At $\mathcal{G} = 35$ (the value corresponding to the high-T quantum critical regime), the spectral function exhibits overdamped behaviour in real time and a double peak structure is present in the frequency domain. Apart from the remnant of the quasiparticle peak at larger frequencies, a smaller contribution at near-zero frequencies anticipates the low-frequency behaviour of the spectral function near the critical point. This double-peak structure indicates an intricate interplay between diffusive modes dominating at the critical point and the quasiparticle remnant from the paramagnetic phase, leading to strongly correlated dynamics [38].

Finally we note that the classical spectral functions obtained here are directly related to the ones in the high-T quantum theory by the matching procedure described in chapter 2 and appendix A: Inserting the renormalized mass $M$ obtained from dimensional reduction leads to the quantum results for $m_R$ and the damping rate as shown in Refs. [40, 44].

**Universal behaviour near $\mathcal{G}_c$.**    At a phase transition, the spectral function exhibits a qualitatively different behaviour by acquiring a universal scaling form [37]

$$\rho(t,\mathbf{p}=0) = t^{\frac{2-\eta}{z}-1} g\left(\frac{t}{\xi_t}\right),\tag{13}$$

with anomalous dimension $\eta$, and dynamic critical exponent $z$, which in principle is independent of static critical exponents near thermal (i.e. classical) phase transitions [51]. The universal scaling function $g$ is expected to behave as

$$g\left(\frac{t}{\xi_t}\right) \sim \exp\left(-\frac{t}{\xi_t}\right),\tag{14}$$

with a diverging temporal correlation length $\xi_t$ at the critical point. With the exactly known static critical exponent [52] $\eta = 0.25$ and the previously found dynamic critical exponent $z = 2$ in this model [37, 53], we expect an algebraic behaviour $\rho(t,\mathbf{p}=0) \sim t^{-1/8}$ and $\rho(\omega,\mathbf{p}=0) \sim \omega^{-7/8}$ for long times and small frequencies at the critical point $\mathcal{G}_c$.

In the insets of Fig. 1, we display the critical spectral function at $\mathcal{G} = 61.5 \approx \mathcal{G}_c$ on a double logarithmic scale, recovering the expected algrebraic behaviour of the spectral function with exponents in rough agreement with the expected values (black dashed lines). As we find no exponential decay on the observed time scales we conclude that $\xi_t \gg 50/M$ for $\mathcal{G} = 61.5$, with the algebraic decay setting in at around $Mt = 10$. Remarkably, we already find near-critical behaviour of the spectral function for $\mathcal{G} = 50$, however, with the exponential decay of $g(t/\xi_t)$ dominating the dynamics. From a fit to an exponential for times $Mt > 20$ we find $\xi_t \approx 30/M$. The oscillatory behaviour before the algebraic/exponential decay is related to short-time high-momentum physics, that are remnants of quasi-particles [37]. At higher momentum (not plotted), we indeed find well-defined quasiparticle peaks even at $\mathcal{G}_c$ with a linear dispersion $\omega = |\mathbf{p}|$, i.e. with a group velocity equal to the speed of light, albeit with drastically reduced weight compared to the zero-frequency, zero-momentum peak.

**Ordered phase $\mathcal{G} > \mathcal{G}_c$.**    In the symmetry broken phase (not plotted) we again find a quasiparticle peak, however with a significantly higher renormalized mass $m_R \approx 3.5M$ and a large

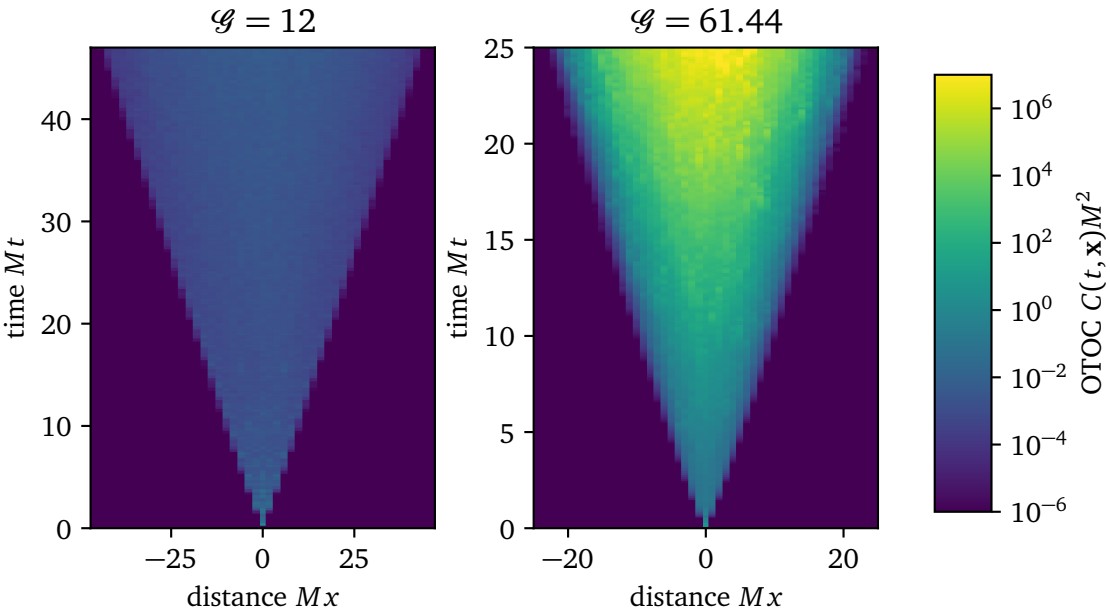

Figure 2: **Ballistic spreading of OTOC.** Cut of the OTOC along one axis. For all values of $\mathcal{G}$ (left: $\mathcal{G} = 12$,, right: $\mathcal{G} = 61.44$) studied in this work, we find ballistic spreading of the OTOC. However, both the butterfly velocity and the Lyapunov exponent associated with the exponential growth within the lightcone shows a strong dependence on $\mathcal{G}$. Due to the ambiguity of the color scale, it is not possible to read off the butterfly velocity directly from this plot. For a detailed discussion see Sec. 4.2.

broadening of $\Gamma \approx 1.5/M$ at $\mathcal{G} = 90$. These gapped excitations correspond to the amplitude fluctuations in the minima of the effective potential. There are no gapless Goldstone modes due to the discrete $\mathbb{Z}_2$ symmetry of the order parameter.

## 4 Many-body chaos

By studying the spectral function we have found qualitatively distinct regimes characterized by the presence of well-defined quasiparticles for small $\mathcal{G}$, a cross-over to a broad spectrum at $\mathcal{G} \approx 35$ and universal, algebraic low-frequency behaviour near $\mathcal{G}_c$ without well-defined quasiparticle excitations at zero momentum. In this section, we study the dependence of the OTOC $C(\mathbf{x}, t)$ on $\mathcal{G}$.

**Ballistic spreading for all $\mathcal{G}$.** In Fig. 2 we display a cut of the OTOC $C(\mathbf{x}, t)$ along one real-space axis for small $\mathcal{G}$ and near $\mathcal{G}_c$, showing ballistic spreading. The same holds for all other values of $\mathcal{G}$ studied in this work. Furthermore, we find perturbations to spread throughout the lightcone. As indicated by the logarithmic color scale, we also find exponential growth of the OTOC within the light-cone, with the Lyapunov exponent $\lambda_L$ associated to that growth being strongly dependent on the fluctuation parameter $\mathcal{G}$. The ballistic spreading needs to be contrasted with the emergent non-relativistic dynamic exponent $z = 2$ found from the spectral function at the phase transition as the OTOC still shows a "$z = 1$" behaviour.

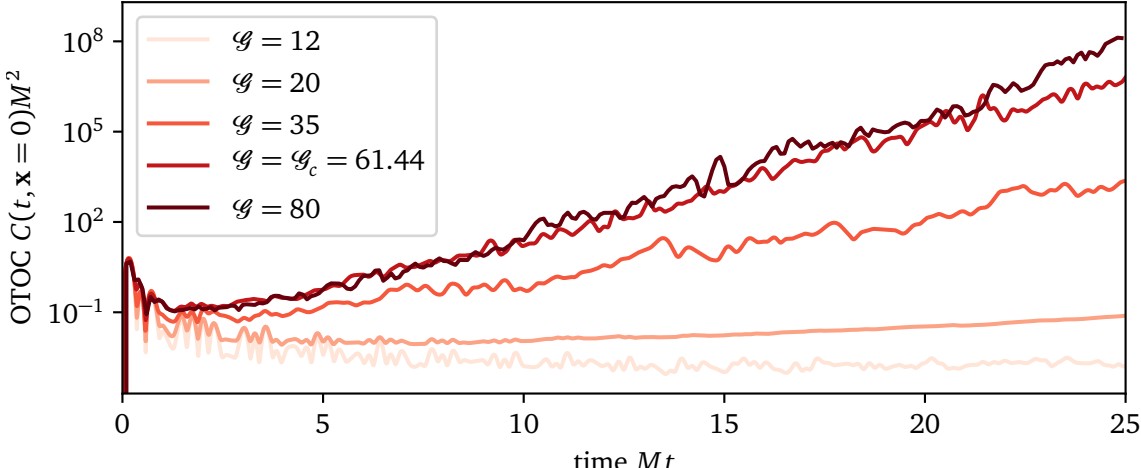

Figure 3: **Local OTOC.** After the trivial short-time dynamics related to the spreading of a sharply peaked momentum-field perturbation, chaotic exponential growth sets in with late time behaviour dominated by the largest Lyapunov exponent, which is strongly dependent on the value of the fluctuation parameter $\mathcal{G}$. For the smallest value $\mathcal{G} = 12$, exponential growth only sets in after the time scale shown here; we hence evolved larger systems to longer times to determine the Lyapunov exponent in that case.

**Absence of critical slowing down in the OTOC.**   In Fig. 3, we show the time evolution of the local OTOC for several values of $\mathcal{G}$. The short time behaviour ($Mt < 2$) is not dependent on $\mathcal{G}$ and is merely related to the initial conditions chosen: The initially non-vanishing momentum perturbation at the origin is converted into a $\delta$-function-like shape of the field perturbation as $\delta\pi = \partial_t \delta\varphi$, which explains why the OTOC $\langle\{\varphi(\mathbf{x}, t), \pi(\mathbf{x}, 0)\}\rangle$ gives the same results as the one studied here. Subsequently, the dynamics is dominated by the large gradient to neighbouring lattice sites, i.e. the Laplacian in the equations of motion (see Eq. (9)) is far larger than the other terms. As soon as the differences to neighbouring lattice sites are washed out, the non-linear dynamics created by the interplay between the term proportional to the mass squared and the one including the coupling to the field dominates. Importantly, we find that for all values of $\mathcal{G}$ these late-time dynamics are given by an exponential growth of $C(\mathbf{x} = 0, t)$, with an approximately constant exponent for late times. This behaviour has to be contrasted with the spectral function, for which we found algebraically slow decay for $Mt \gtrsim 10$ at $\mathcal{G} = \mathcal{G}_c$ (see Fig. 1). For $\mathcal{G} \gtrsim 50$, the temporal correlation length is larger than $Mt = 30$ such that the time scales shown in Fig. 3 are well within the regime showing exponential damping in the spectral function, where the decay rate diverges as the phase transition is approached. The OTOC instead *grows* exponentially with an increasing exponent as $\mathcal{G}$ increases.

To assess these findings more quantitatively, we study the Lyapunov exponent of the exponential growth, the space-time shape of the OTOC, the butterfly velocity associated to the ballistic spreading and fluctuations of the OTOC in the following. We have checked in Appendix C that all our results are independent of both lattice spacing and system size and thus determine the chaotic properties of the continuum field theory in the thermodynamic limit.

## 4.1 Lyapunov exponent

Lyapunov exponents are a standard measure of chaos in classical dynamics [54], quantifying the rate of separation between two neighbouring trajectories. In general, the number of Lyapunov exponents is equal to the number of degrees of freedom, quantifying the rate of change

associated with a perturbation in every direction of phase space. In our case, the number of degrees of freedom is given by $2 \times N^2 \approx 10^5$ for the typical lattice sizes studied here, so that determining the whole Lyapunov spectrum (scaling quadratically with the number of degrees of freedom [55]) is not constructive. We hence focus on the largest Lyapunov exponent $\lambda_L$ (in the following simply called "the Lyapunov exponent") by defining

$$\lambda_L = \lim_{t \to \infty} \frac{1}{2t} \ln C(\mathbf{x} = 0, t). \tag{15}$$

Here the limit of vanishing perturbation is implicit in this definition as we are evaluating the OTOC directly in this limit, c.f. Eq. (9). In practice, we determine the Lyapunov exponent by a fit with an exponential to $C(\mathbf{x} = 0, t)$ at late times. Note that in this semi-classical model the OTOC is not expected to saturate, whereas in a full quantum theory the OTOC is expected to deviate from the semi-classical exponential growth around the Ehrenfest time [42]. Our classical theory may hence be viewed as an effective theory for a quantum system at high temperatures such that the time scales studied here are below the Ehrenfest time of the corresponding quantum theory. Similar conclusions have been drawn previously from the perspective of semi-classical trajectories [42] and for fidelity-OTOCs in the Dicke model [56]: In regimes dominated by classical modes, the exponential growth of the OTOC is described by the classical statistical approximation. Further evidence may be obtained from the large-N, high $T$ calculation in Ref. [16]. There, it was found that the rungs in the Bethe-Salpeter equation for the OTOC contributing to the exponential growth are dominated by classical modes with $\mathbf{p} < T$. This may be understood from the fact that the "Wightman" correlators responsible for the chaotic contributions of those rungs are at $T \gg \omega$ proportional to the correlation functions $\langle \varphi \varphi \rangle$, which are hence strongly peaked at $\omega \approx \sqrt{m_R^2 + \mathbf{p}^2}$ and have a weight strongly decreasing with $\mathbf{p}$ (due to the FDR, c.f. the discussion of the finite $\mathbf{p}$ behaviour of the spectral function in chapter 3). While we determined $\lambda_L$ from the local OTOC at position $\mathbf{x} = 0$, we found exponential growth with the same $\lambda_L$ for all $\mathbf{x}$ as long as the light-cone has passed, i.e. for times $Mt > |M\mathbf{x}|/v_B$, where $v_B$ is the butterfly velocity studied below. This implies that the Lyapunov exponent can equivalently be defined as $\lambda_L = \lim_{t \to \infty} \frac{1}{2t} \ln C(\mathbf{p} = 0, t)$.

**Local, static approximations fail to reproduce exponential growth.** As we are determining the Lyapunov exponent from a local quantity, one could suspect from the equations of motion of the perturbation (see Eq. (9)) that the chaotic exponential growth can be related to an instability of independent anharmonic oscillators on each lattice site with an (in general complex) frequency given in terms of the static value of the volume average of $\tilde{\varphi}^2$. However, in this approximation we have found *stable* oscillations for all values of $\mathcal{G}$ considered here. Hence, they can not reproduce the exponential growth of the OTOC, which is a genuine dynamical many-body effect in this theory.

**Algebraic approach of $\lambda_L$ to noninteracting limit.** In Fig. 4 we show the Lyapunov exponent as defined in Eq. (15) as a function of the fluctuation parameter $\mathcal{G}$. For small $\mathcal{G}$, we find an approximate power law behaviour $\sim \mathcal{G}^2$, where $\mathcal{G} = 0$ is the non-interacting limit in which $\lambda_L = 0$. Algebraic approach of a non-chaotic limit has been previously found[4] in many other classical dynamical models [54, 58].

---

[4]One is inclined to put the recently found algebraic behaviour $T^{0.48}$ of the Lyapunov exponent in a classical spin system [33] more into the perspective of this classical order-to-chaos transition rather than the conjectured connection to the linear-in-$T$ bound in quantum many-body chaos, especially as in many classical dynamical models the exponent was found to be given by the Feigenbaum constant $\approx 0.449$ [57].

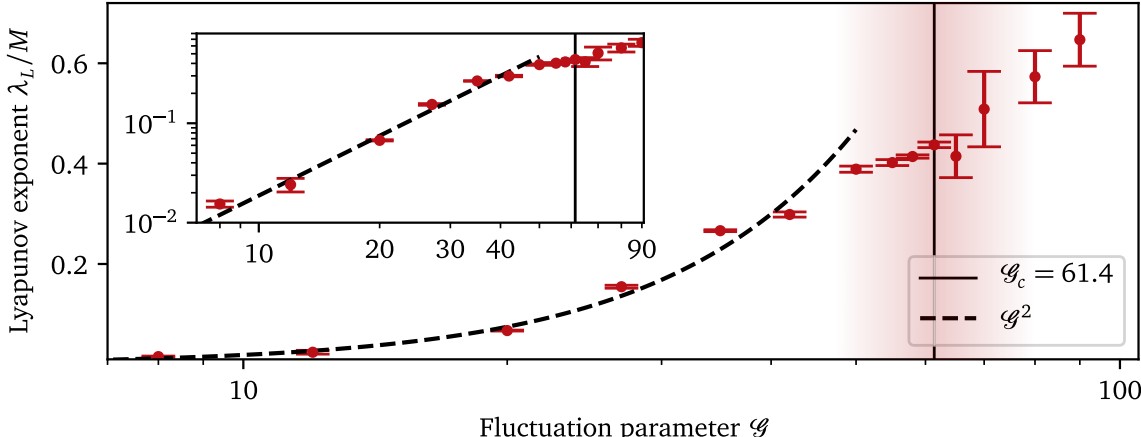

Figure 4: **Lyapunov exponent.** The Lyapunov exponent $\lambda_L$ increases as a function of $\mathcal{G}$, with a different approach to the phase transition from above and below. The red shading indicates the range of $\mathcal{G}$ in which critical slowing down was found in the spectral function. An approximate power-law behaviour $\sim \mathcal{G}^2$ is found for $\mathcal{G} \lesssim 40$ (see inset). Error bars are statistical errors obtained from a jackknife binning analysis detailed in App. C.

**Linear-in-$T$ behaviour in quantum critical regime.** By employing the matching procedure described in App. A we can relate the results found here for the classical theory to the corresponding quantum theory at high temperatures in the quantum critical regime, in which $\mathcal{G} \approx 35$ and $M = \sqrt{2}\pi T/3$ [41]. Inserting our numerical result for $\mathcal{G} \approx 35$ and using that we measure $\lambda_L$ in units of $M$ we get

$$\lambda_L^{\text{quantum crit.}} \approx (0.12 \pm 0.01)\pi T. \tag{16}$$

This linear-in-T scaling was conjectured to be the universal behaviour in strongly correlated quantum systems and in particular, our result is within the conjectured MSS bound[5] $\lambda_L \leq \pi T$ [10]. In a recent large-N $d = 2$ calculation in the $O(N)$ model (for which ours is the $N = 1$ variant) the linear-in-T behaviour was also reproduced, but with a substantially larger prefactor. We note however, that $N = 1$ is special in $d = 2$ due to the presence of a finite-temperature phase transition related to the discrete symmetry of the field. The case of $N = 1, d = 2$ was also shown to be special in far-from-equilibrium phenomena [59] otherwise universal for all $N$ and $d \leq 3$ [60]. It would hence be interesting to study the $N \gg 1$ limit within the classical statistical approximation and compare with the diagrammatic approach.

**Signatures of a crossover near the phase transition.** We find that the $\mathcal{G}^2$ behaviour found for small $\mathcal{G}$ crosses over to a slower increase for large $\mathcal{G}$, with the point of the crossover between the two behaviours approximately corresponding to $\mathcal{G}_c$. Previously, a cusp-like behaviour has been found at the phase transition of a classical XY model [61]. A maximum of the Lyapunov exponent near second-order phase transitions found in some other models [58, 62] has later been attributed [55] to the divergence of the specific heat near the phase transition, i.e. the Lyapunov function to be a smooth function of the energy. We find a similar behaviour in our case (not shown in plot), with the Lyapunov exponent *decreasing* as a function of $\langle H \rangle/VT$, with large $\mathcal{G}$ corresponding to small $\langle H \rangle/VT$, and with no apparent special feature near the energy density corresponding to $\mathcal{G}_c$. Directly at the phase transition we find

$$(\lambda_L/M)\big|_{\mathcal{G}=\mathcal{G}_c} = 0.44 \pm 0.01. \tag{17}$$

---

[5]Note that we define the Lyapunov exponent by a factor of two differently to the authors of the bound.

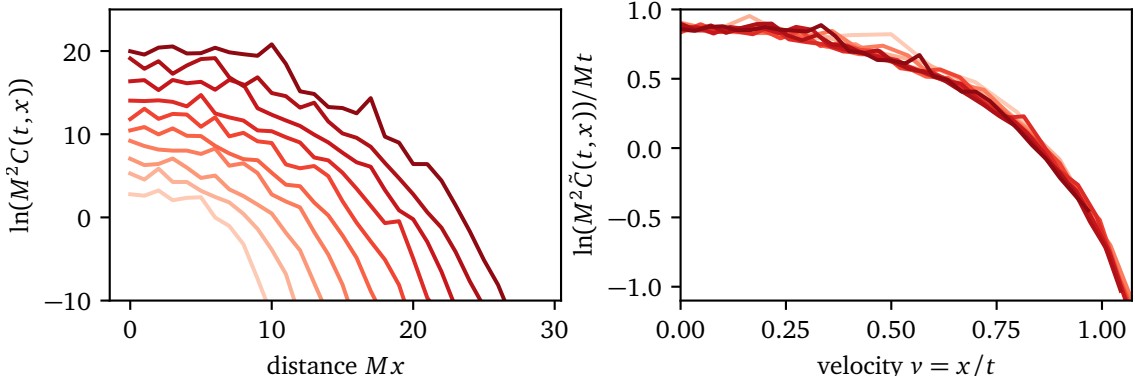

Figure 5: **Scaling collapse of the OTOC.** On the left we display equally spaced time slices in the interval $Mt \in [10, 30]$ for $\mathcal{G} = 61.44$ along one spatial axis, with times increasing from bright to dark colours. The collapse in the right plot indicates that $C(t, x) \sim \exp(\lambda(v)t)$, with a scaling function $\lambda(v) = \lambda(x/t)$. The OTOC $C(t, x) \to 0$ outside the causal lightcone $v/c = 1$ in the limit $a_s \to 0$, which we checked numerically. We plot $\tilde{C}(t, x) = C(t, x) - \text{const.}$, where the constant is the intersection with the y-axis of the linear fit to $\ln(C(t, x = 0))$ for $Mt > 10$.

We note that $M > 0$ in the dimensional reduction scheme discussed here [41] and in particular it stays finite at the phase transition[6].

**Ordered phase $\mathcal{G} > \mathcal{G}_c$.** Even though $\lambda_L$ is numerically difficult to obtain in this region (see App. C), we found $\lambda_L$ to continue to rise for $\mathcal{G} > \mathcal{G}_c$, however somewhat slower than below the phase transition. In order to find more conclusive results in the ordered phase, a description in terms of a "dual" weakly coupled classical field theory [41, 63] might lead to further insights into the chaotic properties of the ordered phase and make numerical simulations considerably easier.

## 4.2 Butterfly velocity

While so far having mainly focussed on the time evolution of the local OTOC, we extend our analysis to the full space-time dependence in the following. After showing that the OTOC follows an approximately self-similar time evolution, we define the butterfly velocity unambiguously via the resulting scaling function. Lastly, we show that the butterfly velocity exhibits a global maximum at the phase transition.

**Space-time dependence of the OTOC.** Cuts of the OTOC along one axis of the 2D spatial plane are shown in Fig. 5 for several times. The scaling collapse obtained on the right side of the plot indicates that the OTOC follows a self-similar time evolution of the form

$$C(\mathbf{x}, t) \sim \exp(\lambda(v)t), \tag{18}$$

with a velocity-dependent Lyapunov exponent $\lambda(v) \equiv \lambda(|\mathbf{x}|/t)$ [25, 64–67] and where the exact functional form of $\lambda(v)$ in general depends on $\mathcal{G}$. We however qualitatively found it to be similar for all values of $\mathcal{G}$. In particular, it smoothly crosses zero at some finite v and hence wave front broadening as obtained in chaotic quantum lattice models is not present here [22, 23, 25, 67]. In order to obtain the scaling collapse, we plot $\tilde{C}(t, x) = C(t, x) - \text{const.}$,

---

[6]It is the mass renormalized by classical statistical fluctuations $m_R$ which vanishes at the phase transition, as visible from the study of the spectral function in chapter 3.

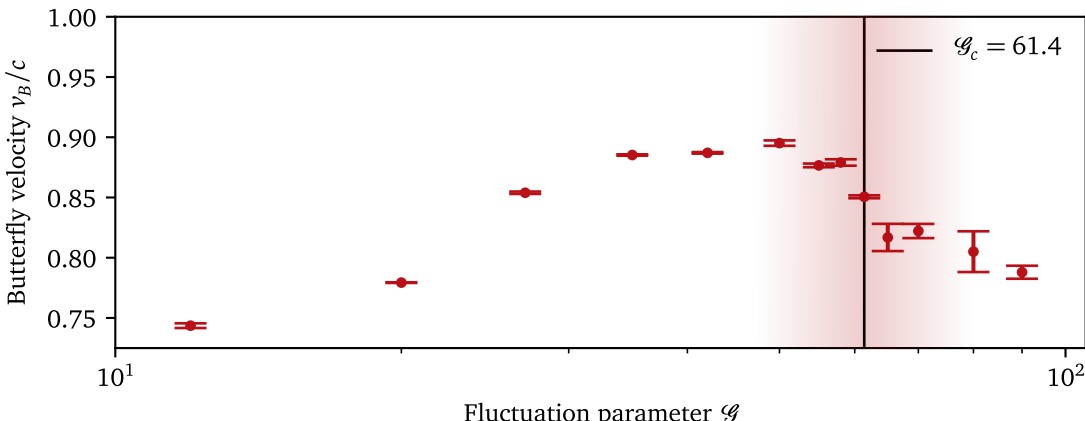

Figure 6: **Butterfly velocity.** The velocity associated to the ballistic spreading of the OTOC exhibits a global maximum on approach to the phase transition at $\approx 0.89c$, and is considerably smaller than the speed of light $c$ for a wide range of values for $\mathcal{G}$. Different behaviour is seen on both sides of the phase transition: While the butterfly velocity is slowly rising upon approaching the phase transition from the paramagnetic phase, it steeply decreases as one moves away from it into the symmetry broken phase. The red shading indicates the range of $\mathcal{G}$ in which critical slowing down was found in the spectral function. Error bars are obtained from a jackknife binning analysis.

where the constant is the intersection with the y-axis of the linear fit to $\ln(C(t, x = 0))$ for $Mt > 10$ and is related to the non-universal time evolution for $Mt < 10$.

**Butterfly velocity.** The above finding of a self-similar evolution of the OTOC in space-time motivates the definition of the butterfly velocity as the velocity $v_B$ at which $\lambda(v)$ crosses zero, i.e.

$$\lambda(v = v_B) = 0, \tag{19}$$

corresponding to the "slice" of constant velocity $v$ at which the OTOC neither grows nor decays with time.

In Fig. 6 we show the butterfly velocity as a function of the fluctuation parameter $\mathcal{G}$. We find that $v_B$ is always significantly smaller than the speed of light $c$. In particular, in the high temperature quantum critical regime, $\mathcal{G} = 35$, of the corresponding quantum field theory we have

$$v_B^{\text{quantum crit.}} = (0.853 \pm 0.001)c, \tag{20}$$

which is significantly smaller than $v_B \approx c$ found in the high-T phase of the O(N) model at large N [16]. As the phase transition is approached from the paramagnetic phase, the butterfly velocity saturates around $\mathcal{G} = 50$ at the maximum value of approximately $0.89c$ before dropping sharply just at the phase transition as one moves into the symmetry broken phase.

The maximum of the butterfly velocity and hence maximally fast spreading of OTOCs at the phase transition is in sharp contrast to the diffusively slow order parameter dynamics found in the spectral function. This shows the qualitatively different behaviour of many-body chaos and transport and shows that the former is a new, in general independent measure of thermalization. We note however, that $\lambda_L$ and $v_B$ might not be completely independent parameters as a universal relation between the diffusion constant, $\lambda_L$ and $v_B$ has been conjectured [11, 17].

## 4.3 Fluctuations

Previous studies in quantum lattice models [22, 23, 25] showed that the growth of the $(D-1)$ dimensional OTOC *front* can be mapped to interface growth in the $(D-1)$ KPZ universality class (in 1D, diffusive behaviour of the front is found). Classical spin chains in 1D [34], however, found indications that the effective equations of motion of the OTOC itsself follows KPZ universality, i.e. that its run-to-run fluctuations have a self-similar behaviour with universal exponents. In a (2+1)D classical spin model, no such behaviour was however found [33].

Here, we study the run-to-run fluctuations of the time-dependent height variable

$$h(t) = \ln\left(\{\varphi(\mathbf{x} = \mathbf{0}, t), \varphi(\mathbf{0}, 0)\}_{PB}^2\right)/2, \tag{21}$$

which is the single-run generalization of $\ln(C(t)/2)$. Therefore, the slope of $h(t)$ is the single-run generalization of the Lyapunov exponent $\lambda_L$. In general, however, $\langle h(t)\rangle_{\text{cl}} \neq \ln(C(t)/2)$ due to the non-commutativity of sample average and logarithm (see Ref. [68]). We study the time evolution of the probability distribution of $h(t)$ to quantify fluctuations of the local OTOC.

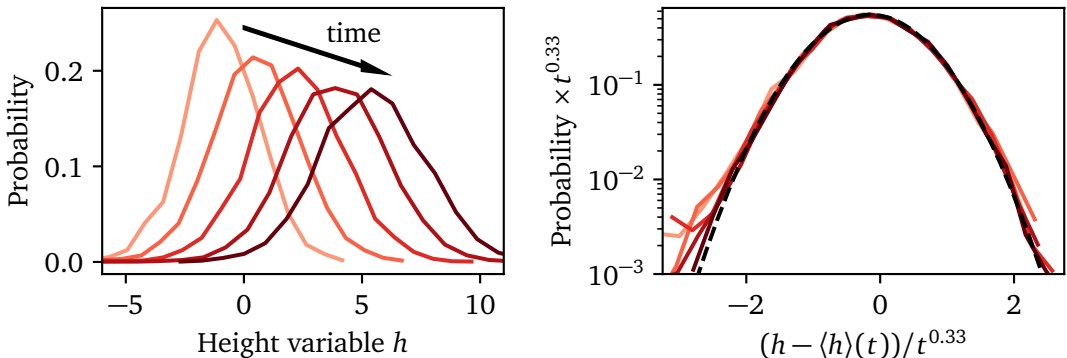

Figure 7: **Self-similarity in the run-to-run fluctuations of the OTOC.** Time dependent probability distribution of the height variable $h(t)$ defined in the main text at $\mathcal{G} = 61.44$, near the phase transition. All curves collapse after a rescaling with $t^\alpha$, $\alpha = 0.33 \pm 0.06$ . The scaling function is close to a Gaussian (dashed black line), with residual deviations from scaling and Gaussianity vanishing as time progresses. Times increase from bright to dark lines and are given by $Mt \in \{11, 16, 21, 26, 30\}$.

In Fig. 7 we show that the probability distribution $P(h, t)$ follows the (in time) self-similar scaling form

$$P(h, t) = t^{-\alpha} P_S\left(\frac{h - \langle h\rangle(t)}{t^\alpha}\right), \tag{22}$$

where we determined

$$\alpha = 0.33 \pm 0.06 \tag{23}$$

from a $\chi^2$ analysis, see appendix E. This value of $\alpha$ is in accordance with the (1+1)D KPZ universality class and hence the scenario of a fluctuating OTOC front also found in quantum lattice models [22, 23, 25]. We however can not fully exclude the OTOC itsself fluctuating, as found in 1D classical spin chains [34], as (2+1)D KPZ behaviour with exponent 0.24 is within 1.5 standard deviations of our result.

The scaling distribution $P_S$ closely follows a Gaussian with mean $-0.16$ and variance $0.77$. Note that in the regime accessible here, a Gaussian is not distinguishable from a Tracy-Widom distribution expected from KPZ universality [69].

**Short time behaviour.** Times earlier than $Mt \approx 3$ deviate substantially from the self-similar scaling form in Eq. 22, exhibiting large (negative) skewness as well as a shift of the maximum towards larger $h$, corresponding to the local maximum in the time evolution visible in Fig. 3. In the range $3 < Mt < 10$, the distribution is still substantially skewed, but approximately coincides with the scaling form around the maximum. A leftover of this evolution is visible in the rescaled plot in Fig. 7 for times $Mt > 10$, where deviations in the tails of the distribution from the scaling form become smaller as time progresses.

# 5   Discussion and Outlook

In this work, many-body chaos in the self-interacting $\lambda\varphi^4$ real scalar field theory has been discussed at high temperatures and near its second order thermal phase transition. By employing dimensional reduction we reduced this quantum field theory to a classical statistical field theory and argued why dynamical and chaotic properties may be captured by this approximation. Subsequently, we employed a numerical method motivated from linear-response theory to study the classical equivalent of the out-of-time-ordered correlator (OTOC) and used the classical fluctuation-dissipation relations to study the spectral function.

Opposed to the diffusive order parameter transport near the phase transition we found ballistic spreading of OTOCs in the whole parameter regime. The Lyapunov exponent exhibits the linear-in-T behaviour conjectured to be universal in the quantum critical regime. Furthermore, we found some indications of a different functional form of the approach to the critical fluctuation strength on both sides of the phase transition. Most importantly, we found the butterfly velocity to have a global maximum near the phase transition, indicating that OTOCs spread quickest in this strongly correlated regime. We contrasted these findings with the order parameter dynamics from the spectral function and argued how many-body chaos offers an independent characterization of thermalization dynamics.

While temporal fluctuations of the OTOC were found to be consistent with the KPZ universality class, further investigations would be necessary to fully confirm this. Especially spatial fluctuations could be studied, as well as the dependence of the exponents on the dimensionality.

While we argued here that many aspects of quantum many-body chaos can be captured by the classical statistical approximation and reproduced many results from diagrammatic quantum field theory calculations, it would be advantageous to test these assumptions within a unified framework. A possible route to do so would be to use a Bethe-Salpeter equation on a doubled Keldysh contour obtained from the two-particle-irreducible effective action [70] in some analogy to the calculation of the shear viscosity [71] as classical and quantum contributions can be naturally identified in this formalism [72]. This could also offer a route to benchmarking the non-perturbative approximations (such as the $1/N$ expansion) frequently employed to diagrammatically studying OTOCs in field theories [16, 17, 39].

Furthermore, it would be interesting to investigate the connection of chaos to the hydrodynamic modes, in this model the transverse momentum, a diffusive mode, and the pressure, a damped ballistic mode. It has been previously shown that the diffusion constant stays finite at the phase transition [53]. It would hence be interesting to study whether the conjectured connection [11] $D = v_B^2/\lambda_L$ between the diffusion constant $D$, the Lyapunov exponent and the butterfly velocity also holds in this model as previously found in other classical models [33] and quantum field theories [17].

Our method of obtaining the OTOC can be straightforwardly applied to other field theories. As the classical statistical approximation is valid in highly occupied regimes, it would be suited to studying chaos in ultracold Bose gases in the condensed phase. Another application could

be cases in which high occupations are present out of equilibrium, such as in the early stages after a heavy-ion collision as described by classical Yang-Mills theory [73].

## Acknowledgements

We thankfully acknowledge insightful discussions with Jürgen Berges, Avijit Das, Sarang Gopalakrishnan, Sergej Moroz, Asier Piñeiro Orioli, Tibor Rakovszky, and Félix Rose.

**Funding information** A.S. acknowledges financial support from the International Max Planck Research School for Quantum Science and Technology (IMPRS-QST) funded by the Max Planck Gesellschaft (MPG). We furthermore acknowledge support from the Technical University of Munich - Institute for Advanced Study, funded by the German Excellence Initiative and the European Union FP7 under grant agreement 291763 and from the DFG grant No. KN1254/1-1, DFG TRR80 (Project F8). This work was also funded by the Deutsche Forschungsgemeinschaft (DFG, German Research Foundation) under Germany's Excellence Strategy – EXC-2111 – 390814868.

## A   Dimensional reduction

We are interested in a real scalar quantum field theory at finite temperature described by the partition function

$$Z = \int \mathcal{D}\phi(x,\tau)\exp(-S), \tag{24}$$

with Euclidean action

$$S[\phi] = \int_0^{1/T} \mathrm{d}\tau \int \mathrm{d}^2\mathbf{x} \left\{ \frac{1}{2}(\partial_\tau \phi)^2 + \frac{1}{2}(\nabla_{\mathbf{x}}\phi)^2 + \frac{1}{2}(r_c + r)\phi^2 + \frac{g}{4!}\phi^4 \right\}. \tag{25}$$

Above we have introduced the deviation $r$ of the quadratic coupling from the (quantum) critical coupling $r_c$ and the quartic coupling $g$.

In the following, we summarize the procedure of dimensional reduction, in which the above $(2+1)$ dimensional quantum field theory can at high temperatures and near the finite temperature phase transition be reduced to the two dimensional classical statistical field theory discussed in the main text. To do so, we mainly follow the line of argumentation in Refs. [36, 41, 74].

**Free theory.**   To motivate the general procedure, consider for a moment $g = 0$. Introducing the fields $\phi(\tau,\mathbf{x}) = \sum_{n\in\mathbb{R}} e^{i\omega_n \tau}\phi_n(\mathbf{x})$ in Matsubara space, the action becomes

$$S^{U=0}[\phi] = \frac{1}{T}\sum_n \int \mathrm{d}^2\mathbf{x} \left\{ \frac{1}{2}(\nabla_{\mathbf{x}}\phi_n)^2 + \frac{1}{2}(r_c + r + 4\pi^2 n^2 T^2)\phi_n^2 \right\}. \tag{26}$$

At high temperature, the masses of the non-zero Matsubara modes become very large. As these determine the inverse correlation length, the low momentum properties are entirely dominated by the $n = 0$ mode. Hence, all modes with $n \neq 0$ may be omitted and the theory is described by a two dimensional classical statistical theory involving only the $n = 0$ mode.

**Interacting theory.** For $g \neq 0$, it is expected that the non-zero Matsubara modes have thermal masses of order of at least $T$. The same procedure may hence be followed in cases in which the $n = 0$ mode has a thermal mass smaller than $T$, which is especially true near a thermal phase transition [74]. Then, we can perturbatively integrate out all non-zero Matsubara modes, replacing the bare couplings in the action by renormalized ones in an expansion with respect to the in general small mass of the $n = 0$ mode in units of temperature. We denote these by $r + r_c \to m^2$ and $g \to \lambda$. In order to render the thermal mass of the zero mode smaller than $T$, we use the results from Ref. [41]. There, a $\epsilon = 3 - d$ expansion was used, resulting in $m^2 \sim \epsilon T^2 \ll T^2$. After removing the only UV divergence in this theory by introducing a one loop renormalized mass $M^2$ according to Eq. 4 in the main text, the couplings to be inserted into the classical effective theory result to lowest order as

$$M^2 = \left( \frac{\sqrt{2}}{3} \pi T \right)^2, \tag{27}$$

$$\mathcal{G} = 8\sqrt{2}\pi \approx 35.5 \tag{28}$$

in the high-T quantum critical regime in $d = 2$.

**Dynamics.** While these arguments are strictly only valid for static properties, it has been shown that low frequency, low momentum dynamical properties such as the damping rate of the quasiparticle in the spectral function yields the same result in quantum and classical field theory after matching only *static* quantities [44, 45]. To do so, a canonically conjugate field momentum term $\pi = \partial_t \varphi$ needs to be introduced as done in Eq. (1). Although it is natural to choose the same form of the field momentum in the classical theory as in the quantum theory, this is not a unique choice as dynamical and static properties are independent in classical field theory [51]. In appendix B we give a complementary discussion of the range of validity of the classical statistical approximation for the field dynamics from the perspective of the equilibrium limit of non-equilibrium quantum field theory.

# B  Range of validity of the classical-statistical approximation

Classical statistical field theory is a good approximation for quantum field theory in regimes in which the statistical fluctuations, given by the anticommutator of fields $F = \left\langle \frac{1}{2}\{\hat{\varphi}, \hat{\varphi}\} \right\rangle - \langle \hat{\varphi} \rangle^2$, are much larger than the quantum fluctuations, given by the commutator of fields (i.e. the spectral function) $\rho = i \langle [\hat{\varphi}, \hat{\varphi}] \rangle$,

$$|F(t_1, t_2; \mathbf{x}_1, \mathbf{x}_2)| \gg |\rho(t_1, t_2; \mathbf{x}_1, \mathbf{x}_2)|. \tag{29}$$

This classicality condition can be motivated from a slightly more restrictive, but rigorous condition derived from two-particle irreducible effective action methods in (non-)relativistic scalar field theories [39, 72] and is applicable both in equilibrium and far-from-equilibrium. In thermal equilibrium, $F$ and $\rho$ are only dependent on relative coordinates $t_1 - t_2$ and $\mathbf{x}_1 - \mathbf{x}_2$ and are linked by the fluctuation-dissipation relations in temporal frequency space, $F = -i(1/2 + n_T)\rho$, where $n_T(\omega) = 1/(\exp(\omega/T) - 1)$ is the Bose-Einstein distribution. Although the classicality condition in Eq. (29) must strictly be fulfilled for all $(t_1, t_2; \mathbf{x}_1, \mathbf{x}_2)$, one may argue that a theory already behaves classically if it is dominated by modes fulfilling this condition. Consequently, a theory in thermal equilibrium behaves classically if the spectral function is dominated by momentum modes $\mathbf{p}$ with energy $\omega \ll T$ as then $n_T(\omega) \approx T/\omega \gg 1$, i.e. the Bose-Einstein distribution reduces to the classical Rayleigh-Jeans law.

In the presence of well-defined quasiparticles, $\rho$ is strongly peaked at frequency $\omega \approx \sqrt{m^2 + \mathbf{p}^2}$ (neglecting the momentum dependence of the effective mass). At zero momentum, Eq. (29) can therefore then be specified to the condition

$$m^2 \ll T^2, \tag{30}$$

which coincides with the condition for dimensional reduction discussed in appendix A.

Moreover, directly at the phase transition $\mathcal{G} = \mathcal{G}_c$, the classicality condition above is exactly fullfilled as the zero-momentum spectral function diverges as $\omega \to 0$. This justifies why zero-momentum, zero-frequency properties of quantum field theories at finite temperature phase transitions are rigorously described by classical field theory [37].

## C  Numerical implementation

**Discretization.** The Hamiltonian in Eq. 1 in terms of the rescaled variables discretized on an $N \times N$ square lattice with rescaled lattice spacing $\tilde{a}_s = a_s M$ is given by

$$H/T = \frac{1}{2} \tilde{a}_s^2 \sum_{\mathbf{x}} \left[ \tilde{\pi}_{\mathbf{x}}^2 - \tilde{\varphi}_{\mathbf{x}} \frac{1}{\tilde{a}_s^2} \sum_{\mathbf{e}_i} \left( \tilde{\varphi}_{\mathbf{x}+\mathbf{e}_i} - 2\tilde{\varphi}_{\mathbf{x}} + \tilde{\varphi}_{\mathbf{x}-\mathbf{e}_i} \right) + \frac{m^2}{M^2} \tilde{\varphi}_{\mathbf{x}}^2 + \frac{\mathcal{G}}{12} \tilde{\varphi}_{\mathbf{x}}^4 \right], \tag{31}$$

where we have partially integrated the gradient term in order to write it in terms of a second order discretization of the resulting Laplacian. In the latter, $\mathbf{e}_i$ denote the lattice unit vectors. The bare mass squared is given in terms of the discretized gap equation (Eq. 4) as

$$\frac{m^2}{M^2} = 1 - \frac{\mathcal{G}}{2V} \sum_{\mathbf{p}} \frac{1}{\mathbf{p}^2 + 1}, \tag{32}$$

where the lattice momenta are given by [70]

$$\mathbf{p}^2 = \sum_{i=1}^{2} \frac{4}{\tilde{a}_s^2} \sin^2 \left( \frac{\pi n_i}{N} \right), \tag{33}$$

with $n_i \in \{0, .., N-1\}$.

We use a leapfrog discretization with time step $d\tilde{t} = dt M$ for the equations of motion of the field,

$$\frac{\tilde{\varphi}_{\mathbf{x}}(t+1) - 2\tilde{\varphi}_{\mathbf{x}}(t) + \tilde{\varphi}_{\mathbf{x}}(t-1)}{d\tilde{t}^2}$$
$$= \left[ \frac{1}{\tilde{a}_s^2} \sum_{\mathbf{e}_i} \left( \tilde{\varphi}_{\mathbf{x}+\mathbf{e}_i}(t) - 2\tilde{\varphi}_{\mathbf{x}}(t) + \tilde{\varphi}_{\mathbf{x}-\mathbf{e}_i}(t) \right) - \frac{m^2}{M^2} \tilde{\varphi}_{\mathbf{x}}(t) - \frac{\mathcal{G}}{6} \tilde{\varphi}_{\mathbf{x}}^3(t) \right], \tag{34}$$

as well as for the equations of motion of the field perturbation,

$$\delta\tilde{\varphi}_{\mathbf{x}}(t+1) = 2\delta\tilde{\varphi}_{\mathbf{x}}(t) - \delta\tilde{\varphi}_{\mathbf{x}}(t-1) + d\tilde{t}^2 \times$$
$$\left[ \frac{1}{\tilde{a}_s^2} \sum_{\mathbf{e}_i} \left( \delta\tilde{\varphi}_{\mathbf{x}+\mathbf{e}_i}(t) - 2\delta\tilde{\varphi}_{\mathbf{x}}(t) + \delta\tilde{\varphi}_{\mathbf{x}-\mathbf{e}_i}(t) \right) - \frac{m^2}{M^2} \delta\tilde{\varphi}_{\mathbf{x}}(t) - \frac{\mathcal{G}}{2} \tilde{\varphi}_{\mathbf{x}}^2(t) \delta\tilde{\varphi}_{\mathbf{x}}(t) \right]. \tag{35}$$

The initial state of the fields are given by Monte Carlo sampling as described below whereas we initialize the perturbation of the momenta as

$$\delta\tilde{\pi}_{\mathbf{x}}(0) = c\delta_{\mathbf{x0}}, \tag{36}$$

where $c$ is a random number drawn uniformly from the interval $[-0.1, 0.1]$ and the field perturbation $\delta\tilde{\varphi}_{\mathbf{x}}(0) = 0$. We have checked that the results for the Lyapunov exponent and the butterfly velocity do not depend on the choice of interval.

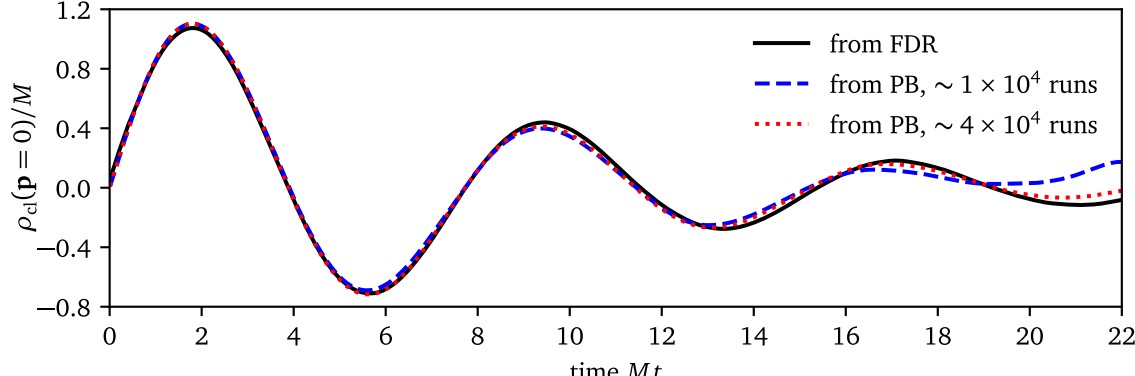

Figure 8: **Comparison of two different numerical approaches to obtain the spectral function.** Spectral function at zero momentum for $\mathcal{G} = 20$ as obtained from the fluctuation-dissipation relations (FDR) (Eq. (7)) and the Poisson Bracket (PB) (Eq. (8)), where the former is fully converged with respect to the number of runs. The latter converges to the FDR result as the number of runs is increased. Deviations at early times are due to the finite time step (see text).

**Discretized OTOC.** As we need to prepare an initial state for the perturbation $\sim \delta(\mathbf{x})$ in the continuum, and therefore $\sim \frac{1}{\tilde{a}_s^d} \delta_{\mathbf{x0}}$ on the d-dimensional lattice, the lattice spacing dependence is given as

$$\left( \frac{\delta \tilde{\varphi}(t, \mathbf{y})}{\delta \tilde{\pi}(0, \mathbf{x})} \right)^2 \rightarrow \tilde{a}_s^{2d} \left( \frac{\delta \tilde{\varphi}_{\mathbf{y}}(t)}{\delta \tilde{\pi}_{\mathbf{x}}(0)} \right)^2, \tag{37}$$

i.e. the results obtained with the initial state in Eq. (36) have to be *divided* by $\tilde{a}_s^{2d}$ to obtain results independent of the lattice spacing. We have furthermore tested that using double or quadruple computer precision does not result in a considerable difference.

**Spectral function from FDR and PB.** In Fig. 8 we compare the spectral function as obtained from the fluctuation dissipation relation (Eq. (7)) and the Poisson bracket (Eq. (8)). Both methods converge for intermediate and late times as the number of runs is increased. Deviations occur at early times as the spectral function obtained from the FDR, opposed to the one from the PB, does not exactly vanish at initial time due to the finite time step [38].

**Critical spectral function.** In order to obtain enough statistics to evaluate the critical spectral function in Fig. 1, we exploit time-translational invariance [37] of thermal equilibrium by transforming to Wigner coordinates $\rho(t_1, t_2) = \rho(T = \frac{1}{2}(t_1 + t_2), \tau = t_1 - t_2)$ and averaging over the 'center of mass time' $T$. To transform to Fourier space with respect to the relative coordinate $\tau$, we used the antisymmetry of $\rho$ to perform the discrete sine transform

$$\rho(t, \mathbf{p} = 0) = \frac{a_s^d}{N^d} \sum_{\mathbf{x}} \left( 2 \int d\tau \rho_{\mathbf{x}}(\tau) \sin(\omega \tau) \right), \tag{38}$$

where $d$ is the spatial dimension. We have also used a Gaussian filter to reduce finite-time oscillations in the Fourier transformed spectra, but have checked that the overall behaviour is robust against change of filter window.

**Hybrid Monte Carlo.** We start the Monte Carlo algorithm by initializing the fields in a state drawn from the thermal state of the free theory ($\mathcal{G} = 0$) with a Gaussian of zero mean and standard deviation $1/\sqrt{(\tilde{a}_s(m^2/M^2))^2 + d}$, with $d = 2$ being the spatial dimension [70]. This ini-

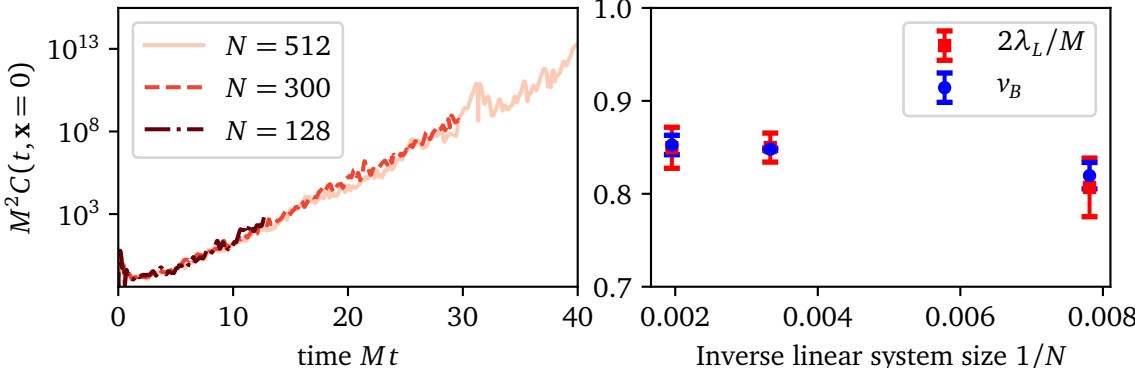

Figure 9: **Convergence of the OTOC at $\mathcal{G} = 61.5$ with system size.** Both the convergence of the local OTOC (left) as well as butterfly velocity and lyapunov exponent (right) with respect to system size indicate that there are no considerable finite size effects for $N = 300$, the system size chosen for most plots in the main text. We stopped the simulations at times $t \sim N a_s$ to avoid boundary effects.

tialization has lead a much shorter thermalization time compared to initialization with a standard deviation independent of $\mathcal{G}$. We then iterate the following Hybrid/Hamiltonian Monte Carlo (HMC) step introduced in the context of lattice QCD [75, 76].

1. Draw some initial $\tilde{\pi}(\mathbf{x})$ from a Gaussian distribution with zero mean and standard deviation $1/\sqrt{\tilde{a}_s}$. Evaluate the Hamiltonian, giving the energy $E_1/T$.

2. Evolve this state in time using Eq. 34 with step size $d\tilde{t} = \epsilon$ for a number of time steps $N_t$.

3. Evaluate the Hamiltonian again, giving $E_2/T$.

4. Accept the new configuration with probability $\min(1, \exp(-(E_2/T - E_1/T))$.

We have usually chosen $\epsilon \approx 0.01\tilde{a}_s$ and $\epsilon N_t \approx 1$, yet large values of $\mathcal{G}$ required slightly smaller $\epsilon$. Furthermore, the number of time steps was randomized by $\pm 10\%$ in order to circumvent possible periodicities of the trajectories. We started the measurement runs from a pre-equilibrated state obtained after approximately $1000 - 5000$ Monte Carlo steps (thermalization as monitored from the convergence of the energy was usually reached after $\approx 50$ steps for small $\mathcal{G}$ and took longer for larger $\mathcal{G}$ due to critical slowing down). Measurements were taken after approximately $30 - 50$ steps, where the autocorrelation of the fields with the initial (pre-equilibrated) state was negligible. This turned out not to be the case in the symmetry broken phase due to the difficulty of escaping one of two deep minima of the effective potential. We hence used independently thermalized MC initial conditions with approx 5000 Monte Carlo steps. Furthermore, we checked for several values of $\mathcal{G}$ that we averaged over enough realizations (usually $10^3 - 10^4$) by checking convergence of Jackknife errors for the butterfly velocity and Lyapunov exponent from a binning analysis [77].

**Convergence with system size and lattice spacing.** In Figs. 10,9 we compare the time evolution of the local OTOC $C(\mathbf{x} = 0, t)$ as well as the butterfly velocity and Lyapunov exponent for different lattice spacings and system sizes. No considerable dependency on those two parameters is seen around the conventionally chosen ones ($Ma_s = 0.2, N = 300$). We find a tendency for larger differences between different lattice spacings than between different system sizes, indicating that the OTOC and its properties have a stronger dependence on the

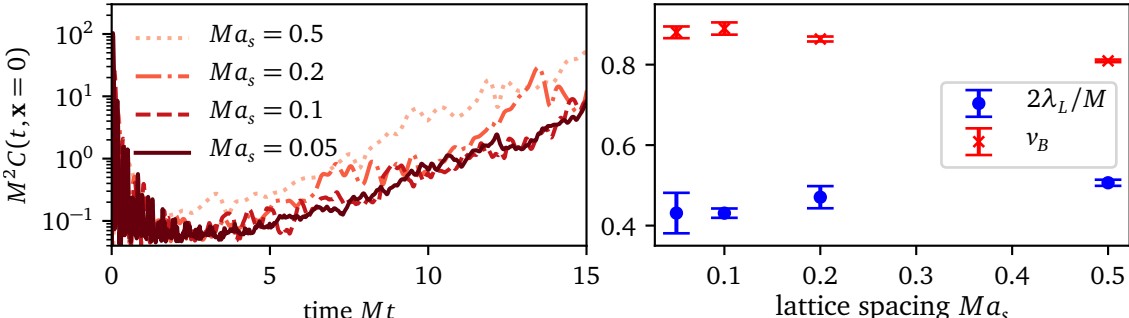

Figure 10: **Convergence of the out-of-time ordered correlator with lattice spacing.** (left) Although quantitative differences between lattice spacings are visible in the out-of-time correlator at $\mathcal{G} = 35$, they show the same qualitative behaviour with lattice spacings $Ma_s \leq 0.1$ oscillating around an exponential growth. The oscillations become smaller when increasing the number of Monte Carlo runs (not shown). (right) Butterfly velocity and Lyapunov exponent as a function of lattice spacing. Both quantities are independent of lattice spacing within error bars at around $Ma_s = 0.2$, the value chosen for all simulations shown in the main text.

UV than the infrared cut-off. This also holds near the phase transition, since the OTOC is not sensitive to critical slowing down.

**Jackknife binning analysis.** In order to estimate errors and convergence with respect to sample size, we employed a jackknife binning analysis. Dividing up the ensemble of samples into $M$ blocks of size $k$, it proceeds by calculating an observable $\mathcal{O}$ (such as the Lyapunov exponent) of an ensemble-average in which one of those blocks has been omitted. Denoting $\langle \mathcal{O} \rangle$ the observable calculated on the whole ensemble and $\langle \mathcal{O} \rangle_i$ the one where block $i$ has been removed before averaging, the jackknife estimate for the mean and standard deviation are given by [78]

$$\langle \mathcal{O} \rangle_{JK} = \langle \mathcal{O} \rangle - (M-1)\left( \frac{1}{M} \sum_{i=1}^{M} \langle \mathcal{O} \rangle_i - \langle \mathcal{O} \rangle \right), \tag{39}$$

$$\Delta \mathcal{O} = \sqrt{\frac{M-1}{M} \sum_{i=1}^{M} \left( \langle \mathcal{O} \rangle_i - \frac{1}{M} \sum_{i=1}^{M} \langle \mathcal{O} \rangle_i \right)}. \tag{40}$$

In order to check convergence of the errors with respect to the sample size, the standard deviation needs to be plotted as a function $k$ for a fixed sample size. If the standard deviation converges, then so has the sample average.

## D   Phase transition

The model considered in this work (c.f. Eq. (1)) exhibits an equilibrium phase transition from a symmetric phase with $\langle \varphi \rangle = 0$ to a symmetry broken phase with $\langle \varphi \rangle \neq 0$. Due to our renormalization procedure, the critical value of $\mathcal{G}$ does not strongly depend on the lattice spacing and can be found by standard finite size scaling procedures at a small enough, but fixed $a_s$. Here, we study the Binder cumulant $B_4$ [79] given by

$$B_4 = 1 - \frac{\langle \varphi^4 \rangle}{3 \langle \varphi^2 \rangle^2}, \tag{41}$$

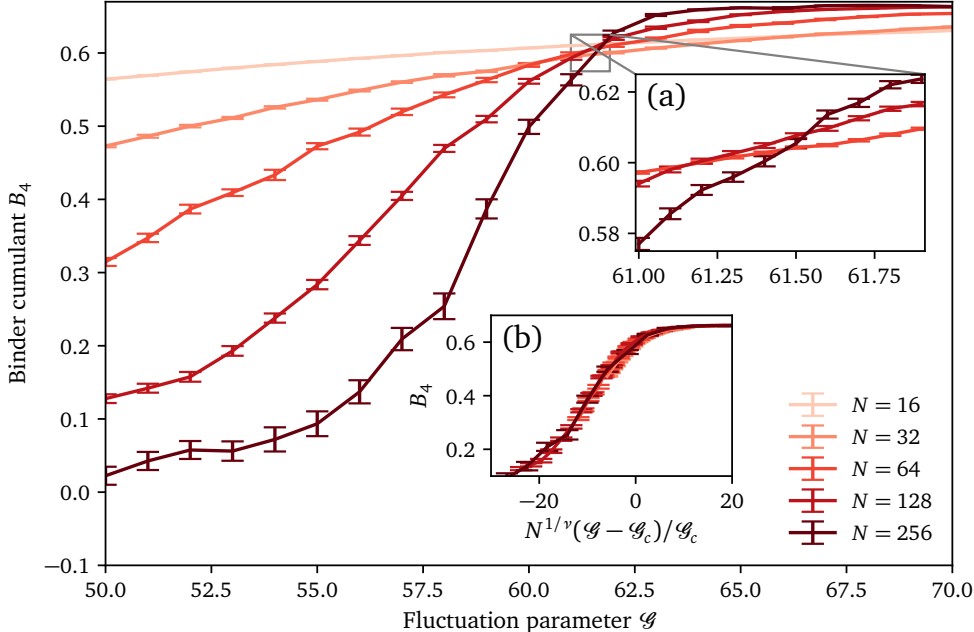

Figure 11: **Phase transition.** Binder cumulant as defined in Eq. (41) as a function of the fluctuation parameter $\mathcal{G}$ for different system sizes, where the inflexion point as determined from the largest system sizes in inset (a) reveals the critical fluctuation parameter to be $\mathcal{G}_c \approx 61.4$ in agreement to previous studies [47]. Rescaling all curves as shown in inset (b) with the known critical exponent $\nu = 1$ for the 2D Ising universality class leads to a good collapse over a large parameter regime. Lines connecting data points are solely a guide for the eyes. Error bars are obtained from a jackknife binning analysis.

with limits $B_4 \to \frac{2}{3}$ for $\mathcal{G} \to \infty$ and $B_4 \to 0$ for $\mathcal{G} \to 0$ and a universal value $B_{4,c} \approx 0.6104$ [80] at the phase transition. In above expression, $\varphi = \frac{1}{N^2} \sum_{\mathbf{x}} \varphi_{\mathbf{x}}$ denotes the volume averaged field.

In Fig. 11 we determine the critical Binder cumulant $B_{4,c}$ as well as fluctuation parameter $\mathcal{G}_c$ from the intersection of $B_4(\mathcal{G})$ of the three largest system sizes considered. We find $\mathcal{G}_c = 61.38 \pm 0.16$ and $B_{4,c} = 0.604 \pm 0.004$ consistent with previous results [47, 80] and with errors mainly resulting from residual finite-size effects. Furthermore, we show the finite size collapse of system sizes $N \geq 32$, assuming that $B_4$ is directly a finite size scaling function of the form $B_4 = B_4((\mathcal{G} - \mathcal{G}_c)N^{1/\nu}, \dots)$.

# E  Estimating the exponent of a scaling collapse

In order to estimate the exponents for the scaling collapse of the fluctuations in section 4.3, we performed a $\chi^2$ analysis. First, we change the definition of the scaling ansatz in Eq. 22 slightly by introducing a reference time $t_{\text{ref}}$,

$$P_{\text{resc}}(t, \bar{h}) = \left(\frac{t}{t_{\text{ref}}}\right)^{-\alpha} P\left(t, \left(\frac{t}{t_{\text{ref}}}\right)^{-\alpha} \bar{h}\right), \tag{42}$$

where we also introduced $\bar{h} = h - \langle h \rangle(t)$. Usually, we chose $M t_{\text{ref}} = 11$, i.e. the beginning of the regime in which almost no deviation from self-similar behaviour is visible (see Sct. 4.3 for details). According to this definition, a perfect scaling collapse would correspond to

$$\Delta P = P_{\text{resc}}(t, \bar{h}) - P(t_{\text{ref}}, \bar{h}) = 0 \quad \forall t \in \text{scaling regime.} \tag{43}$$

In order to quantify deviations from scaling, we define the error function

$$\chi^2(\alpha) = \frac{1}{N_t} \sum_{t=t_{\text{ref}}}^{t_{\text{max}}} \frac{1}{\int d\bar{h}} \int d\bar{h} \left( \frac{\Delta P(t,\bar{h})}{P(t_{\text{ref}},\bar{h})} \right)^2, \tag{44}$$

where $N_t$ is the number of times in the interval $[t_{\text{ref}}, t_{\text{max}}]$, in our case usually $M t_{\text{max}} = 30$ and $N_t = 20$. Integrals over $\bar{h}$ were numerically evaluated with the trapezoidal rule as in our case $P(t,\bar{h})$ is given in terms of discrete bins from a histogram. We furthermore interpolated $P(t_{\text{ref}},\bar{h})$ linearly to evaluate it at the arguments of $P_{\text{resc}}(t,\bar{h})$.

Finally, we minimized $\chi^2(\alpha)$ to get the most likely value $\bar{\alpha}$ of the scaling exponent. The error is then given by the standard deviation $\sigma$ of the corresponding likelihood function, i.e. $\sigma = \sqrt{\chi^2(\bar{\alpha})}$.

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
