# Peer review of "Many-body chaos near a thermal phase transition"

_SciPost Physics, doi:SciPost Phys. 7, 022 (2019)_

## Round 1 · Referee Report · Anonymous · 2019-6-14

Report
Schuckert and Knap study OTOC growth in a relativistic scalar field theory close to a thermal phase transition using the classical statistical approximation. The properties of OTOCs in field theories and the connection between chaotic classical dynamics and the corresponding quantum dynamics are currently of great interest. The study is very carefully executed and well presented. The only weakness that one could identify is the lack of rigorous arguments as to why the classical statistical simulations faithfully capture the OTOC dynamics of the quantum system. As the authors also discuss, it would be desirable to have a more rigorous argument why the semi-classical method used here, gives faithful results for the quantum mechanical OTOC, or even a direct quantum calculation of it, is desirable but beyond the scope of this work. After discretization the model boils down to an array of classical coupled anharmonic (quartic) oscillators. In my opinion, even the classical dynamics of this system itself is interesting and the authors give good arguments why the classical statistical approximation should be justified.
In conclusion I recommend publication after minor revisions and clarifications (see comments below).
Detailed comments/questions:
1) Formulation “chaos spreads ballistically” is found in several places. I find this a bit imprecise. What spreads is the OTOC, or the support of some Heisenberg operators. Chaos I would regard as a property of the dynamics itself which can be regular or chaotic but not as something that spreads in space.
2) The authors study a classical field theory with classical statistical methods. The argument is then always that this classical theory describes the low low-wavelength, long-distance properties of a corresponding quantum field theory well especially close to the phase transition. Has this conjecture been tested in any way? One argument for the classical description providing meaningful results for the OTOCs in the quantum model is that semi-classical methods have been shown recently to capture the behavior of the OTOC [41]. Are the methods used in [41] equivalent to the ones used here for times smaller than the Ehrenfest time? What is the Ehrenfest time in the model studied here? Later the authors also consider values of G that are rather far away from the phase transition. Especially the quasi-particle peaks studied in Sec. 3 only exist sufficiently far away from the transition.
3) On the quantum versus classical field theory: Sometimes the authors speak of a classical statistical field theory. Although I know that this term is frequently used in the context of statistical mechanics of continuous systems it felt like mixing the term classical field theory with the method that is used to solve/sample it.
4) Appendix A: What is meant by “near” the finite temperature phase transition? Can this be quantified somehow?
5) Last sentence of Sec. 2: All the data shown is for a perturbation in the momentum field only. What exactly is meant by the statement that perturbing in phi gives “similar results” and “same chaotic properties”? Is the statement only valid for the local OTOC or is the equal spatial argument x a typo?
6) P. 6: “Fitting the line shape…” The peaks in Fig 1 do not seems to nicely fit a Breit-Wigner function for G>20. What region was fitted and what are the resulting confidence intervals of the fit?
7) I don’t quite understand in what sense eq. 15 is universal. It seems to me that both sides of the equation are general functions of t. What would a scaling collapse look like?
8) Exponential growth of the OTOC: From Fig. 3 it is not so convincing that the OTOC grows exponentially. It could also be something super-exponential according to the figure. Did the authors calculate longer times to verify the exponential behavior? I imagine that this is hard due to finite size effects. Also the method of obtaining errors for the fitted exponents seems questionable (cf Figure 4): Why would varying the size of the fit interval quantify the quality/confidence of the fit? It is interesting that the authors call this long-time behavior as usually the regime below the Ehrenfest time is considered short time.
In this context I was also wondering why the largest Lyapunov exponent should be the one that governs the OTOC growth. Shouldn’t it rather be the stability of phi with respect to the perturbation in pi which would be two specific phase space directions?
9) The authors mention on page 10 that the OTOC is expected to saturate around the Ehrenfest time. Usually saturation is expected on the Heisenberg time scale. Ref. 41 only argues that the OTOC saturates at times much larger than the Ehrenfest time as far as I understand. (see also e.g. arXiv:1812.09237 [quant-ph])
10) Appendix C “critical spectral function”: Is it correct that the individual trajectories are not time translation invariant?
11) Figure 9: Was the convergence also checked for the non-local OTOCs?
Minor comments:
12) Word doubling “the the” on the bottom of p. 8.
13) Notation: Is could be confusing that lambda is used as the coupling constant as well as Lyapunov exponent and Lyapunov function.
14) In eq 28 and in the text above it the authors use “U” for the interaction parameters while it was defined a “g” before.
15) Eq 39: Should it be 0 instead of x in the denominator?
Author: Alexander Schuckert on 2019-06-23 [id 545]
(in reply to Report 1 on 2019-06-14)
" ": Referee, our reply is always preceded with REPLY
"Schuckert and Knap study OTOC growth in a relativistic scalar field theory close to a thermal phase transition using the classical statistical approximation. The properties of OTOCs in field theories and the connection between chaotic classical dynamics and the corresponding quantum dynamics are currently of great interest. The study is very carefully executed and well presented. The only weakness that one could identify is the lack of rigorous arguments as to why the classical statistical simulations faithfully capture the OTOC dynamics of the quantum system. As the authors also discuss, it would be desirable to have a more rigorous argument why the semi-classical method used here, gives faithful results for the quantum mechanical OTOC, or even a direct quantum calculation of it, is desirable but beyond the scope of this work. After discretization the model boils down to an array of classical coupled anharmonic (quartic) oscillators. In my opinion, even the classical dynamics of this system itself is interesting and the authors give good arguments why the classical statistical approximation should be justified.
In conclusion I recommend publication after minor revisions and clarifications (see comments below)."
REPLY : We would like to thank the referee for the careful reading of the manuscript and the comments. We respond to the questions and comments in detail below and have updated the manuscript accordingly. The reference numbers in this response refer to the old version of the paper (v1 on the arxiv).
"1) Formulation “chaos spreads ballistically” is found in several places. I find this a bit imprecise. What spreads is the OTOC, or the support of some Heisenberg operators. Chaos I would regard as a property of the dynamics itself which can be regular or chaotic but not as something that spreads in space."
REPLY: This is indeed a slightly imprecise formulation. We replaced all occurrences of this formulation with “OTOCs spread ballistically”.
"2) The authors study a classical field theory with classical statistical methods. The argument is then always that this classical theory describes the low-wavelength, long-distance properties of a corresponding quantum field theory well especially close to the phase transition. Has this conjecture been tested in any way?"
REPLY: That critical properties of quantum field theory are rigorously described by a classical field theory at a thermal phase transition is due to the fact that the spectral function diverges for frequency→0 as discussed in Ref.[36]. This argument may be extended to dynamical properties with non-zero frequency by the arguments presented in App. B, i.e. that this is the case if occupations are much larger than the spectral weight. This has been tested in the context of non-equilibrium quantum field theory, where occupations can be tuned independently of the spectral function (Ref.[38]). However, this has not been explicitly shown yet for the OTOC although the results of Ref.[41] indicate that the classical theory captures the correct short-time behaviour (before the Ehrenfest time) in the regimes in which it is expected to be valid by the arguments in App. B.
"One argument for the classical description providing meaningful results for the OTOCs in the quantum model is that semi-classical methods have been shown recently to capture the behavior of the OTOC [41]. Are the methods used in [41] equivalent to the ones used here for times smaller than the Ehrenfest time? What is the Ehrenfest time in the model studied here? Later the authors also consider values of G that are rather far away from the phase transition. Especially the quasi-particle peaks studied in Sec. 3 only exist sufficiently far away from the transition."
REPLY: The first order term of the method employed in [41] (sometimes called the “truncated Wigner approximation”) is indeed very similar to the one we employ, with the only difference being the ensemble over which averaging is done – in TWA quantum fluctuations are incorporated by averaging over a Gaussian ensemble, in our case thermal fluctuations dominate and hence averaging over the Boltzmann weight is employed. Also see arxiv:1906.06143 for a more explicit calculation showing the equivalence of the first order term in Ref.[41] to the TWA. The Ehrenfest time is proportional to the logarithm of the inverse of some effective “hbar”, an example for which is the inverse mean occupation in Bose systems (as employed in Ref.[41]). In our case, the occupations are roughly given by the structure factor F, which is very large in our studied regime at high temperatures T due to the fluctuation dissipation relation which implies F~T (see App. B) . Even though the values of G are far away from the phase transition, the temperature in the corresponding quantum field theory is still considered to be large, such that the classical statistical approximation is still valid (in the dimensional reduction scheme we employ, G and the mass M are independent parameters, which are both functions of the temperature). See e.g. Ref.[37] in which the small G, high T classical statistical field theory results are compared with perturbation theory.
"3) On the quantum versus classical field theory: Sometimes the authors speak of a classical statistical field theory. Although I know that this term is frequently used in the context of statistical mechanics of continuous systems it felt like mixing the term classical field theory with the method that is used to solve/sample it."
REPLY: As the referee mentions the term classical statistical field theory is commonly used. In a way, the word “statistical” just emphasizes that sampling over some ensemble is done, which in our case is the Boltzmann weight.
"4) Appendix A: What is meant by “near” the finite temperature phase transition? Can this be quantified somehow?"
REPLY: Near the phase transition, the thermal mass $m_T$, which is just the inverse correlation length and hence shows some $\sim |T-Tc|^\nu$ behaviour, vanishes at the phase transition, such that $m_T/T=0$ and the dimensional reduction scheme is justified (see further discussion in Zinn-Justin’s book Ref.[72] which we now cite at this point in our manuscript.). “Near the phase transition” may be more specifically defined as the regime in which scaling behaviour is found, for example we find such signatures in the spectral function in Fig.1 already for G=50. This is the way we quantify “near the phase transition”, which we also indicated by the red shading in Figs. 4+6.
"5) Last sentence of Sec. 2: All the data shown is for a perturbation in the momentum field only. What exactly is meant by the statement that perturbing in phi gives “similar results” and “same chaotic properties”? Is the statement only valid for the local OTOC or is the equal spatial argument x a typo?"
REPLY: Thanks for pointing out the typo. The x coordinate in the second component should be set to 0. We have checked for a few values of G that the same Lyapunov Exponent and butterfly velocity results from this slightly changed OTOC. We explain in ch. 4, “absence of critical slowing down” why both these OTOCs are expected to give the same results as the initial condition with a perturbation in the momentum is quickly converted into one with a perturbation in the field.
"6) P. 6: “Fitting the line shape…” The peaks in Fig 1 do not seems to nicely fit a Breit-Wigner function for G>20. What region was fitted and what are the resulting confidence intervals of the fit?"
REPLY: Although the covariances of the fit parameters are vanishingly small in our fit range, about 1e-4%, we agree with the referee that the full shape of G=35 does not fully justify such a fit and hence we removed the fit parameters.
"7) I don’t quite understand in what sense eq. 15 is universal. It seems to me that both sides of the equation are general functions of t. What would a scaling collapse look like?"
REPLY: It is universal in the sense that this scaling form is not dependent on the details of the interactions, but solely on the (dynamical) universality class. A scaling collapse can be obtained by rescaling time with the appropriate temperature scaling of the temporal correlation length, see Fig. 4 in Ref.[36]. The scaling function itself is a function of the rescaled time.
"8) Exponential growth of the OTOC: From Fig. 3 it is not so convincing that the OTOC grows exponentially. It could also be something super-exponential according to the figure. Did the authors calculate longer times to verify the exponential behavior? I imagine that this is hard due to finite size effects. Also the method of obtaining errors for the fitted exponents seems questionable (cf Figure 4): Why would varying the size of the fit interval quantify the quality/confidence of the fit? It is interesting that the authors call this long-time behavior as usually the regime below the Ehrenfest time is considered short time."
REPLY: While we cannot exclude a different functional form from our data, the exponential fit is physically motivated and also found in other studies of the OTOC in classical models (see e.g. Ref.[33]). We have calculated to longer times for N=512 (see Fig.9) and no deviation from the exponential behaviour is found. We also evolved to times larger than speed of light*system size for small G (G=8,12; N=512; Mt<100) and found no deviation from exponential behaviour (no plot shown in paper). We have re-evaluated our error bars in Figs. 4&6 with a Jackknife binning analysis, which also slightly changed the mean values due to bias correction. We have detailed this procedure in App. C. “Long times” is meant here as long compared to the trivial oscillatory short time behaviour for Mt<3.
"In this context I was also wondering why the largest Lyapunov exponent should be the one that governs the OTOC growth. Shouldn’t it rather be the stability of phi with respect to the perturbation in pi which would be two specific phase space directions?"
REPLY: Our analysis differs to the usual Lyapunov stability analysis in that we do not probe the evolution of an infinitesimally displaced trajectory for a short time (to then probe the stability again at the next time instance of the unperturbed trajectory) but rather follow this perturbed trajectory for a large time. As we expect all modes to be coupled in this non-linear system, we expect that the maximally unstable mode will then dominate the growth of all modes for large times. Hence we expect the growth of the OTOC (at least for large times) to be dominated by that mode and hence the largest Lyapunov exponent. A trivial example of this effect is the equivalence of field momentum perturbation and field perturbation discussed above.
"9) The authors mention on page 10 that the OTOC is expected to saturate around the Ehrenfest time. Usually saturation is expected on the Heisenberg time scale. Ref. 41 only argues that the OTOC saturates at times much larger than the Ehrenfest time as far as I understand. (see also e.g. arXiv:1812.09237 [quant-ph])"
REPLY: This is indeed true. For our purposes the time scale which is relevant to us is the one where deviations from the exponential growth are found, which is expected around the Ehrenfest time. We have changed the discussion accordingly.
"10) Appendix C “critical spectral function”: Is it correct that the individual trajectories are not time translation invariant?"
REPLY: We use the time-translational invariance of the individual field trajectories and hence the dependence on only the relative time of the two-point-function to average over the center of mass time of the two-point function as we describe in that section. This is a procedure already used in Ref.[36].
"11) Figure 9: Was the convergence also checked for the non-local OTOCs?"
REPLY: The finite size flow of both the butterfly velocity and the Lyapunov exponent together with the scaling collapse shown in Fig.5 provides strong points for convergence. Checking the convergence of the precise form of lambda(v) is numerically challenging and is beyond the scope of what can be done with our current approach. That is also why we did not attempt to extract the exact functional form of lambda(v).
Minor comments: "13) Notation: It could be confusing that lambda is used as the coupling constant as well as Lyapunov exponent and Lyapunov function."
REPLY: While this is an unfortunate overuse of a single letter, we have tried to stick to the conventions which are currently mostly used in literature. We hope that the subscripts make it clear which quantity is meant.
"12) Word doubling “the the” on the bottom of p. 8.In eq 28 and in the text above it the authors use “U” for the interaction parameters while it was defined a “g” before. Eq 39: Should it be 0 instead of x in the denominator?"
REPLY: Thank you for pointing these typos out, we corrected them.
List of changes in response to referee's comments:
1) Re-evaluated data points in Figs.4+6 with a jackknife binning analysis and removed a couple of points around G_c to increase visibility. New paragraph in App. C explaining the error analysis. Slightly changed the interpretation of the results in 4.1 “Crossover near.”→”Signatures of a crossover near the phase transition” and “ordered phase”, also the corresponding discussion in the conclusions. 2) “Chaos spreads ballistically”->“OTOCs spread ballistically” everywhere 3) Typos indicated by the referee (see above). 4) Chapter3: Removed fit parameters to Breit-Wigner in discussion of G=35. 5) Discussion of Ehrenfest time on p.10 changed 6) Changed second sentence of second paragraph in introduction and added Ref.8. 7) Now referring to new Ref.68 in chapter 4.3
Author: Alexander Schuckert on 2019-07-11 [id 557]
(in reply to Report 2 on 2019-06-24)"This is a well-executed and well-written paper on a topic of great current interest. Almost all computations of OTOCs in quantum system involve some large-N limit, so it is welcome to see an analysis at finite N in a system in finite dimensions. The classical approximation should be qualitatively valid in the quantum-critical region. "
REPLY: We thank the referee for the positive feedback on our work.
"I would ask the authors to comment a bit more on issues related to other works:
(i) They observe a ballistic spread of chaos, unlike the diffusive behavior seen in random unitary networks. Do they have any comment on the reason for this difference ?"
REPLY: Both in our system and in random unitary circuits (RUCs) OTOCs spread ballistically. In RUCs or in noisy spin chains, the operator wavefront broadens diffusively and can be represented by a biased random walk (Refs. [22,23,25]). In these systems different velocity scales have been identified for “information spreading”; the entanglement velocity (measured from the entanglement entropy) and the butterfly velocity (extracted from the OTOC). The discrepancy between the two leads to a finite diffusive spreading of the wave front. It has been observed that the two velocities approach each other when the operator diffusion constant goes to zero (which happens for example when the local Hilbert space dimension ([22,23]) or the noise (Ref.[25]) becomes large). Our model is a field theory which may be identified with a quantum system that locally possesses many degrees of freedom. Therefore our model operates in a different limit than these strongly quantum RUCs or noisy spin chains and may be put into perspective with similar quantum field theory results in large N and weak coupling limits such as the ones presented in references [14-20].
"(ii) Why does 1+1 D KPZ behavior appear in a 2+1 D system ?"
REPLY: In random unitary circuits and deterministic spin chains (see Refs[22,23,25]) it was found for D>=2 that the (D-1) dimensional front of the OTOC behaves according to (D-1) dimensional KPZ universality . In classical models, the situation is however less clear as indications were found that the OTOC itself follows 1D KPZ universality (Ref. [34]) in 1D, whereas less clear results were found in 2D (Ref.[33]). We have re-evaluated and adapted our discussion in chapter 4.3 about fluctuations of the OTOC, indicating that our results give some evidence for (D-1) KPZ universality as in quantum spin chains but emphasizing that we can not unambiguously exclude D-dimensional KPZ universality.

---

## Round 1 · Referee Report · Anonymous · 2019-6-24

Report
This is a well-executed and well-written paper on a topic of great current interest. Almost all computations of OTOCs in quantum system involve some large-N limit, so it is welcome to see an analysis at finite N in a system in finite dimensions. The classical approximation should be qualitatively valid in the quantum-critical region. I would ask the authors to comment a bit more on issues related to other works:
(i) They observe a ballistic spread of chaos, unlike the diffusive behavior seen in random unitary networks. Do they have any comment on the reason for this difference ?
(ii) Why does 1+1 D KPZ behavior appear in a 2+1 D system ?

---

## Round 3 · Referee Report · Anonymous (Referee 1) · 2019-8-7

Report

The authors have resolved all the issues I had raised in my previous report and revised the manuscript accordingly. The paper can now be published without further changes.

---

## Round 3 · List of Changes

In response to Referee 1: 1) Re-evaluated data points in Figs.4+6 with a jackknife binning analysis and removed a couple of points around G_c to increase visibility. New paragraph in App. C explaining the error analysis. Slightly changed the interpretation of the results in 4.1 “Crossover near.”→”Signatures of a crossover near the phase transition” and “ordered phase”, also the corresponding discussion in the conclusions. 2) “Chaos spreads ballistically”->“OTOCs spread ballistically” everywhere 3) Typos indicated by the referee (see answer to report). 4) Chapter3: Removed fit parameters to Breit-Wigner in discussion of G=35. 5) Discussion of Ehrenfest time on p.10 changed 6) Changed second sentence of second paragraph in introduction and added Ref.8. 7) Now referring to new Ref.68 in chapter 4.3

In response to referee 2: 8) Changed discussion of results of fluctuations in chapter 4.3 and corresponding section in conclusions.

---

## Editorial Decision

published